# Factor XIIIA—expressing inflammatory monocytes promote lung squamous cancer through fibrin cross-linking

Alessandro Porrello[1], Patrick L. Leslie[1], Emily B. Harrison[2], Balachandra K. Gorentla[1], Sravya Kattula[3], Subrata K. Ghosh[1], Salma H. Azam[4], Alisha Holtzhausen[1], Yvonne L. Chao[5], Michele C. Hayward[1], Trent A. Waugh[1], Sanggyu Bae[5], Virginia Godfrey[3], Scott H. Randell[6,7], Cecilia Oderup[8], Liza Makowski [1,9,10], Jared Weiss[1,5,11], Matthew D. Wilkerson[12], D. Neil Hayes[1,5,11], H. Shelton Earp[1,11,13], Albert S. Baldwin[1,14], Alisa S. Wolberg[3] & Chad V. Pecot [1,5,11]

Lung cancer is the leading cause of cancer-related deaths worldwide, and lung squamous carcinomas (LUSC) represent about 30% of cases. Molecular aberrations in lung adeno-carcinomas have allowed for effective targeted treatments, but corresponding therapeutic advances in LUSC have not materialized. However, immune checkpoint inhibitors in sub-populations of LUSC patients have led to exciting responses. Using computational analyses of The Cancer Genome Atlas, we identified a subset of LUSC tumors characterized by dense infiltration of inflammatory monocytes (IMs) and poor survival. With novel, immuno-competent metastasis models, we demonstrated that tumor cell derived CCL2-mediated recruitment of IMs is necessary and sufficient for LUSC metastasis. Pharmacologic inhibition of IM recruitment had substantial anti-metastatic effects. Notably, we show that IMs highly express Factor XIIIA, which promotes fibrin cross-linking to create a scaffold for LUSC cell invasion and metastases. Consistently, human LUSC samples containing extensive cross-linked fibrin in the microenvironment correlated with poor survival.

[1] UNC Lineberger Comprehensive Cancer Center, University of North Carolina at Chapel Hill, Chapel Hill, NC 27599, USA. [2] Center for Nanotechnology in Drug Delivery, Eshelman School of Pharmacy, University of North Carolina at Chapel Hill, Chapel Hill, NC 27599, USA. [3] Department of Pathology and Laboratory Medicine, University of North Carolina at Chapel Hill, Chapel Hill, NC 27599, USA. [4] Curriculum in Genetics and Molecular Biology, University of North Carolina at Chapel Hill, Chapel Hill, NC 27599, USA. [5] Division of Hematology & Oncology, University of North Carolina at Chapel Hill, Chapel Hill, NC 27599, USA. [6] Department of Cell Biology and Physiology, University of North Carolina at Chapel Hill, Chapel Hill, NC 27599, USA. [7] Marsico Lung Institute/ Cystic Fibrosis Center, University of North Carolina at Chapel Hill, Chapel Hill, NC 27599, USA. [8] Cancer Immunology, Pfizer, Inc, San Francisco, CA 94080, USA. [9] Department of Nutrition, Gillings School of Global Public Health, University of North Carolina at Chapel Hill, Chapel Hill, NC 27599, USA. [10] Nutrition Obesity Research Center, University of North Carolina at Chapel Hill, Chapel Hill, NC 27599, USA. [11] Department of Medicine, University of North Carolina at Chapel Hill, Chapel Hill, NC 27599, USA. [12] Department of Anatomy, Physiology and Genetics, The American Genome Center, Collaborative Health Initiative Research Program, Uniformed Services University, Bethesda, MD 20814, USA. [13] Department of Pharmacology, University of North Carolina at Chapel Hill, Chapel Hill, NC 27599, USA. [14] Department of Biology, University of North Carolina at Chapel Hill, Chapel Hill, NC 27599, USA. Correspondence and requests for materials should be addressed to C.V.P. (email: pecot@email.unc.edu)

For decades lung cancer has been the leading cause of cancer-related deaths in the U.S. and worldwide[1]. Because non-small cell lung cancer (NSCLC) has a dismal (~15%) 5-year survival rate[2], novel therapies are desperately needed. The recent discovery of select molecular aberrations (e.g. *EGFR* mutations and *ALK* translocations) in lung adenocarcinomas (LUAD) has led to the development of highly effective targeted therapies in these subsets of lung cancer[3]. On the contrary, such advances in the treatment of lung squamous carcinomas (LUSC), which account for about 30% of lung cancer cases, have not materialized. However, the therapeutic blockade of immune checkpoints in LUSC patients has demonstrated exciting responses[4,5]. In fact, several phase III clinical trials recently led to FDA approval of anti-PD1 antibodies in the first- and second-line treatment of LUSC[4–6], suggesting that LUSC may be suitable for additional examination of immune-oncology approaches.

Molecular profiling analyses based on The Cancer Genome Atlas (TCGA) data have revealed that LUSC tumors are highly idiosyncratic and not likely driven by solitary actionable pathways[7]. Using microarray analyses of LUSC tumors, our group previously defined four gene expression subtypes: Classical, Basal, Primitive, and Secretory[8]. These subtypes feature distinct biological processes based on patterns of gene expression. Amongst these subtypes, the Secretory subtype was defined by an immune-response signature rich in genes associated with complement activation, immune cell recruitment, and inflammation[8]. Building upon these observations, we computationally analyzed the LUSC TCGA dataset and identified a new and previously unappreciated subset of LUSC patients that is highly associated with inflammatory monocyte (IM) infiltration and very poor survival.

Tumors recruit IMs ($CCR2^{High}CD14^{+}CD16^{Low}$ in humans; $CCR2^{High}Ly6C^{High}$ in mice) through secretion of the CCL2 chemokine. IMs differentiate into either tumor-associated macrophages (TAMs) or dendritic cells (DCs), and IM-derived TAMs have been intensely investigated for their roles in promoting cancer progression[9–11]. For example, IM-derived TAMs can promote metastasis through production of VEGFa[11,12]. VEGFa has well-recognized roles in distant metastasis formation, in part because it transiently increases vascular permeability to facilitate cancer cell extravasation[12]. TAM secretion of epidermal growth factor (EGF) and IL-6 promote increased migration and "stemness", respectively, of neighboring cancer cells through their paracrine effects in the tumor microenvironment (TME)[13,14]. TAM secretion of IL-10 has pleiotropic roles in immunosuppression through cross-talk with DCs and CD8 + T-cells[15,16]. In agreement with these findings, TAM infiltration into tumors is often associated with poor clinical outcomes in many cancer types[16]. Recently, in opposition to the roles of IMs in cancer, residential monocytes (RMs) ($CX3CR1^{High}CD14^{Low}CD16^{+}$ in humans; $CX3CR1^{High}CD11b^{+}Ly6C^{Low}$ in mice) were found to have inhibitory roles in metastasis formation, largely through scavenging of intravascular cancer cells and recruitment of anti-tumor natural killer T-cells[17]. The divergent roles between IMs and RMs are largely unexplored[18].

Surprisingly, however, little is known about the mechanistic contributions IMs have in metastasis. In fact, IMs are often regarded as inactive precursors in the TME. Additionally, the direct clinical role of IMs in disease progression is largely unknown, particularly in LUSC. Our results have identified a previously unappreciated driver of LUSC metastasis characterized by CCL2-mediated recruitment of IMs and FXIIIA-mediated fibrin cross-linking in the TME, which provides a scaffold for tumor cell invasion. This novel mechanism is reflected in clinical samples where fibrin cross-linking is correlated with poor survival. Thus, IMs in LUSC tumors represent an important context-specific vulnerability of this difficult-to-treat disease.

## Results

**Secretory subtype of LUSC is immune-rich and has poor survival.** Using the LUSC TCGA cohort, we evaluated the survival differences of the intrinsic mRNA subtypes of LUSC[8]. Expanding on results from earlier studies[7,8,19] (Supplementary Fig. 1), we noticed that compared with Classical and Basal subtypes, the Primitive and Secretory subtypes had worse survival; however this was not statistically significant (Fig. 1a, b). Although the subtypes are balanced for clinical stage, hierarchical clustering of RNA-seq data revealed a marked divergence in differentially expressed genes, particularly between the Classical and Secretory subtypes (Fig. 1c, Supplementary Data 1). This corresponded with the Secretory subtype having worse overall survival than Classical (Fig. 1d), most notably among stage II patients (Supplementary Fig. 2a–d). To investigate this divergence, we generated hazard ratios (HR) of survival for the upper portion of the heat map of Fig. 1c (containing over-expressed genes in the Secretory subtype). Of 403 statistically significant "survival genes," we found that Cluster of Differentiation 14 (CD14) was amongst the most significant (Supplementary Data 2). CD14 is a cell surface receptor that is most abundantly expressed on "classical" IMs[20], defined as CCR2$^{High}$CD14+ CD16- cells in humans and CCR2$^{high}$CD11b+ Ly6C$^{High}$ cells in mice[21].

We used Ingenuity Pathway Analysis (IPA) to investigate possible relationships between the "survival genes" and found that leukocyte migration, wound healing, and complement activation emerged as potential tumor-promoting mechanisms within the microenvironment (Fig. 1e, Supplementary Data 3 and 4). Gene ontology (GO) analyses showed that the Secretory subtype is highly enriched for immune-response biological processes, while Classical is characterized by genes of the reduction-oxidation responses (Fig. 1f, Supplementary Data 4). This is consistent with the Classical subtype being highly associated with the KEAP1/NRF2 pathway[7], a tightly coupled antioxidant program, which is also enriched in *KRAS/Lkb1* LUADs that lack immune-response features[22]. Gene Set Enrichment Analysis (GSEA) revealed that top signatures in the Secretory subtype were strongly associated with monocytes and T-cell infiltration (Fig. 1g, Supplementary Data 5). Intriguingly, several markers of IMs, often regarded as precursors of TAMs[23], were leading predictors of LUSC progression, whereas standard macrophage markers displayed more modest survival associations (Fig. 1h).

**Inflammatory monocytes associate with Secretory LUSC tumors.** Splitting CD14 into expression quartiles confirmed a marked separation of the survival curves at the median expression level (Fig. 2a). In fact, CD14 had remarkable prognostic relevance among stage II patients following surgery (Supplementary Fig. 2e), implying its importance in disease recurrence. We found that tumors with above median CD14 expression predominantly enriched for the Basal and Secretory subtypes (Fig. 2b). Consistent with their importance in recruiting IMs and promoting metastasis[11,24], several chemokines (notably CCL2, the classic CCR2 ligand) significantly predicted poor survival and strongly correlated with CD14 expression (Fig. 2c, d). We found an impressive dynamic range of expression for CD14 and these chemokines by subtype, with the Basal and Secretory subtypes consistently having the highest levels (Fig. 2e, f). Recently, CD14 expression in bladder cancer cells was identified as a mechanism of tumor progression[25]. To determine whether LUSC CD14 expression arose from tumor cells and/or tumor-infiltrating leukocytes, we performed immunohistochemistry (IHC) on a LUSC tissue microarray (TMA) previously characterized by subtype (Supplementary Data 6)[8]. Intense membranous CD14 staining was predominately found on tumor-infiltrating leukocytes, and

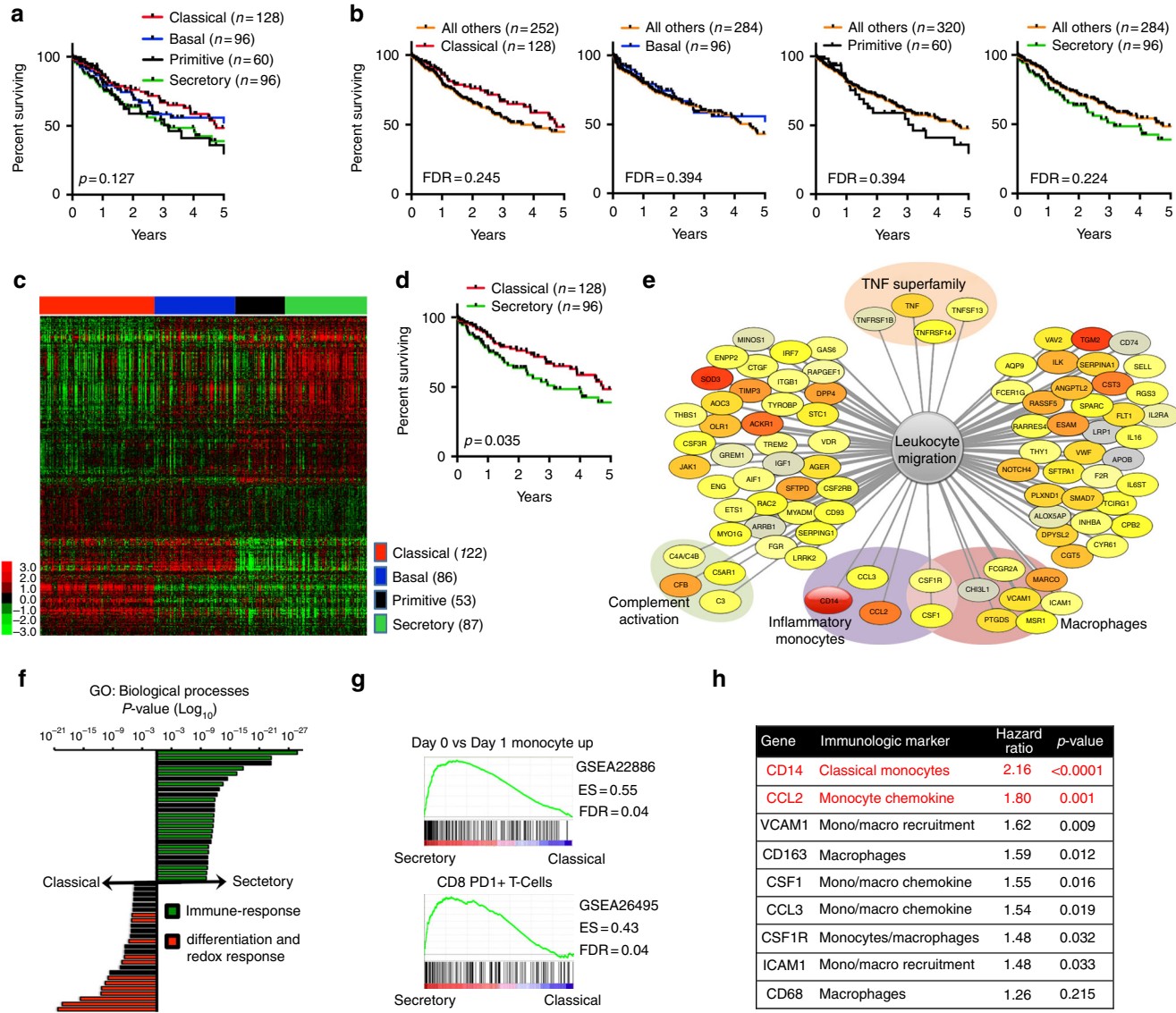

**Fig. 1** Lung squamous carcinoma is associated with inflammatory monocytes. **a** Kaplan–Meier plots of overall survival in lung squamous carcinoma (LUSC) patients ($n = 380$) from The Cancer Genome Atlas (TCGA) categorized by mRNA subtype. **b** Overall survival comparing each mRNA subtype with all others. **c** Hierarchical clustering of ($n = 4291$) genes expressed in 348 patients from the LUSC TCGA RNA-seq dataset. **d** Overall survival comparing Classical versus Secretory subtypes. **e** Network visualization of Ingenuity Pathway Analysis (IPA) of all differentially expressed genes in the upper portion of the heat map (shown in panel **c**) that are statistically significant ($p < 0.05$) in terms of patient survival (in the comparison high ( ≥ median) vs. low ( < median) gene expression), termed "survival genes". (Gray: least significant, Red: most significant) **f**. Gene ontology analysis to investigate the biological processes most linked with genes differentially expressed, moving between the Classical and Secretory subtypes (see the Supplementary Methods for details). Individual bars represent most statistically significant GO terms in either the Classical (red bars) or Secretory (green bars) subtype. **g** Gene Set Enrichment Analyses (GSEA) of the LUSC TCGA dataset. The GSEA is performed going from Secretory to Classical; the GSEA 'mountain plots' show only the two most divergent subtypes. Gene set names were shortened to fit this figure. **h** Table showing 9 genes from the upper portion of the heat map that are associated with reduced overall survival and are markers of monocytes and macrophages. Genes with log-rank p≤0.001 are highlighted in red

rarely on cancer cells (Supplementary Fig. 3). Consistent with RNA-seq data, CD14 protein expression correlated with subtype, and the Secretory subtype displayed the highest levels, followed by Basal (Supplementary Fig. 3). To better characterize the number and location of CCR2$^{High}$CD14+ IMs in these LUSC subtypes, we performed multiplex IHC for CCR2, CD14, and pan-cytokeratin (CK) (Fig. 2g). Across all subtypes, we found that dual positive (CD14+/CCR2+) cells were almost three times more abundant in the stromal (pan-CK negative) regions than in the cancer cell islet (pan-CK positive) regions (Fig. 2g, h). Consistent with mRNA expression levels, there were more dual positive cells in the Basal and Secretory subtypes in both stromal

and cancer cell islet regions (Fig. 2i). To determine what proportion of CD14+/CCR2+ cells represent differentiated M2 macrophages, we performed multiplex IHC for CD14/CCR2/CD206 on 99 lung tumors. This technique revealed that 85% of the immune infiltrates stained exclusively for CD14 and CCR2, suggesting that only a small subset of CD14+/CCR2+ cells represent CD206+ TAMs ($P < 10^{-4}$, Supplementary Fig. 4).

**Inflammatory monocytes correspond with poor survival and CD14.** Although CD14 is most highly expressed on IMs amongst leukocytes[20,26], it is possible that the poor survival associated with

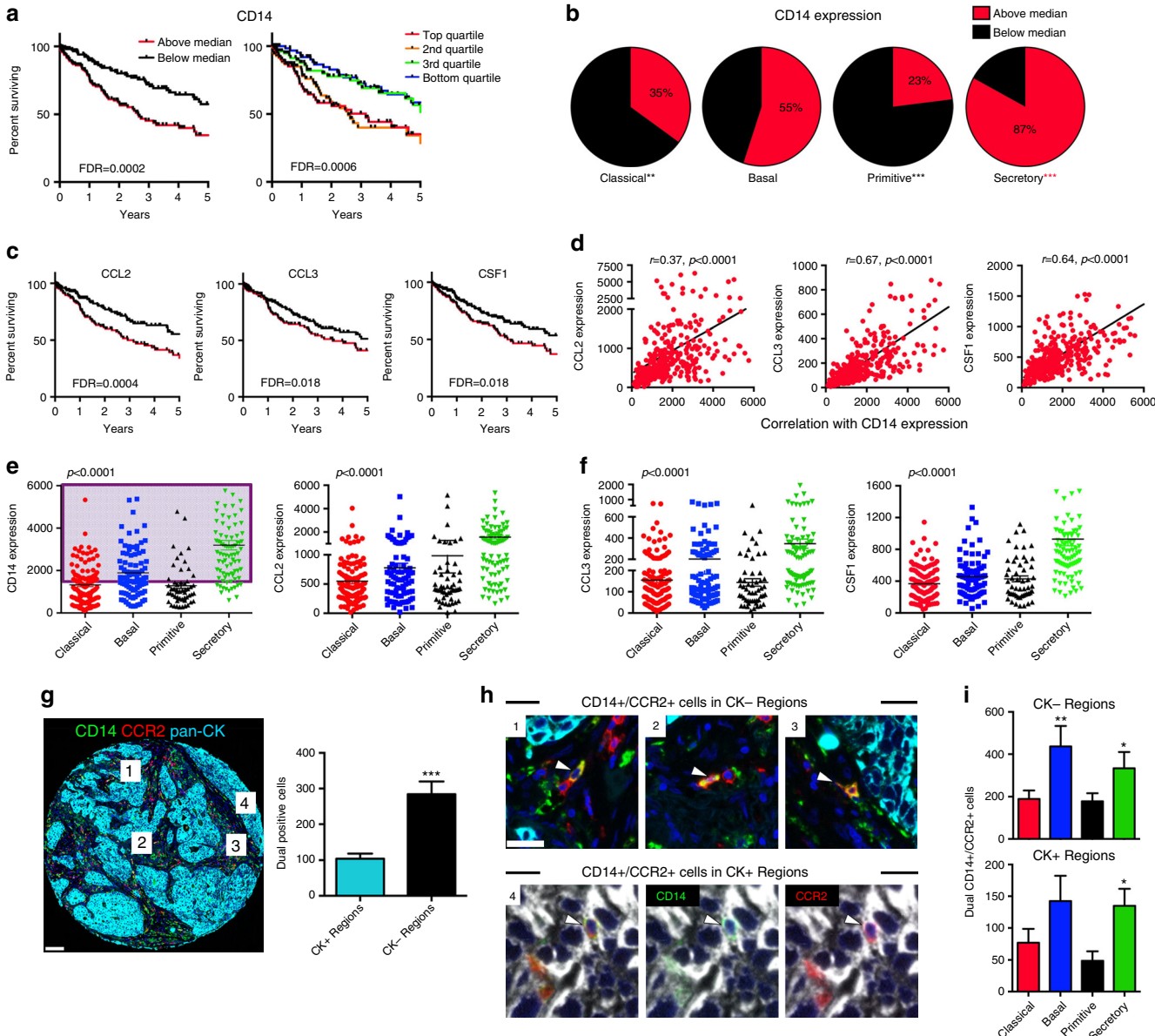

**Fig. 2** Inflammatory monocytes associate with survival in LUSC. **a** Kaplan–Meier plots of overall survival in lung squamous carcinoma (LUSC) patients split by median (left, $p < 0.0001$) and quartile (right, $p = 0.0006$) expression levels of CD14. *P*-values are obtained with the log-rank test; FDR were calculated according to Benjamini and Hochberg. **b** Proportion of patients by mRNA subtype that have CD14 expression levels above (red) or below (black) the median CD14 expression level. Binomial tests for proportions were performed. (Black asterisks: significant enrichment below the median, red asterisks: Significant enrichment above the median). **c** Kaplan–Meier plots of overall survival in LUSC by expression levels of CCL2 ($p = 0.001$), CCL3 ($p = 0.018$) and CSF1 ($p = 0.015$) expression. **d** Pearson's correlations of CCL2, CCL3, and CSF1 chemokines versus CD14 (gene expression). **e** Dynamic range of mRNA expression of CD14 and CCL2 for each LUSC mRNA subtype. *P*-values were obtained with analysis of variance. The purple shading for CD14 expression represents samples in the 'IM-rich subset' (above the median CD14 expression level). **f** Dynamic range of mRNA expression of CCL3 and CSF1 by LUSC mRNA subtype. **g** Representative multiplex IHC for CD14 (green), CCR2 (red) and pan-cytokeratin (light blue) in a LUSC tumor sample, and enumeration of CD14+/CCR2+ cells in CK+ and CK- regions. #1-3 represent CK- regions, #4 represents a CK + region. Scale bar 100 μm. **h** Representative dual CD14+/CCR2+ cells (white arrows) in CK- (#1-3) and CK+ (#4) regions. Note: CK + region shown in white channel to more easily appreciate green and red. Scale bar 25 μm. **i** Enumeration of CD14+/CCR2+ cells in CK- and CK+ regions by mRNA subtype. Classical ($n = 14$), Basal ($n = 9$), Primitive ($n = 6$) and Secretory ($n = 12$). *p*-values were obtained with Student's t-test in comparison to the Classical subtype. * $P < 0.05$, ** $P < 0.01$, *** $P < 0.0001$

high CD14 expression is related to other CD14-expressing immune infiltrates, such as TAMs, DCs, myeloid-derived suppressor cells (MDSCs) or neutrophils[20,26]. We sought to determine whether we could re-classify mRNA subtypes according to the levels of CD14 expression, the degree to which leukocyte cell types correlate with CD14 expression, and how tightly these cell type densities correspond with poor survival.

Given the recent advances in immunogenomic profiling to uncover immune infiltrates in tumors with high fidelity[27,28], we applied a modified 'Immunome' signature across all LUSC TCGA tumors based on a median CD14 expression level threshold (Supplementary Data 7). Classical tumors were predominately represented in the low ( < median) CD14 expression cohort, while Secretory tumors were predominantly in the high ( ≥median)

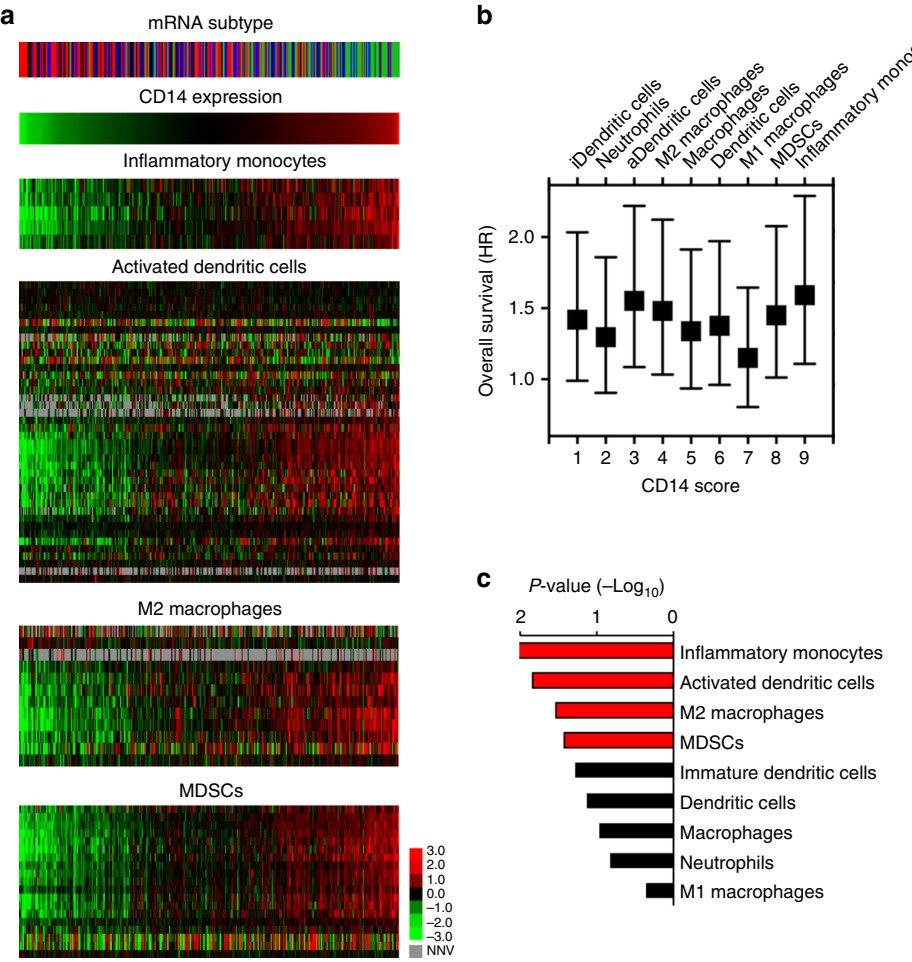

**Fig. 3** IMs have the strongest correlation with CD14 and LUSC survival. **a** At top, the mRNA subtype of each sample is displayed (Classical: red, Basal: blue, Primitive: black, Secretory: green); CD14 expression levels sorting from left (lowest) to right (highest) for the corresponding samples are displayed below. Heat maps of the CD14 + populations that are associated with a statistically significant survival in LUSC are arranged by the most (IMs) to the least (MDSCs) statistically significant. CD14 is a marker both of IMs and MDSCs but is not shown in their heat maps, to avoid data redundancy. Note for the heat map figure legend: gray represents 'null normalized values' (NNV). **b** Scatter plot of overall survival results and CD14 scores for the 9 CD14 + immune cell types. Survival is represented with hazard ratio (HR) ± 95% confidence intervals. Values above 1 indicate worse survival based on cell type density. The CD14 score is the rank of the ratio between the average cell type density score of samples with high (≥ median) vs. low (< median) CD14 levels. **c** Log-rank p-values of overall survival for the 9 CD14 + immune cell types (high vs. low cell density score). The red bars show statistically significant cases, while black bars are used when the statistical significance threshold of 0.05 is not reached

CD14 expression cohort (Fig. 3a, *top*). Basal tumors more often segregated to the high cohort, and Primitive tumors significantly re-classified in the low cohort (Fig. 3a, *top*).

Next, we assessed 9 immune cell types for their correlation with CD14 expression and their individual contribution to overall survival (Fig. 3a, Supplementary Fig. 5). In both analyses, IMs had the strongest relationship with CD14 expression and poor overall survival (Fig. 3b, c). Intriguingly, activated DCs (aDCs) and M2 macrophages, both derivatives of IMs, had the second and third strongest relationships with poor survival, respectively (Fig. 3b, c). These findings imply that IMs have both direct and indirect roles in LUSC progression via differentiation into aDCs or M2 macrophages. Furthermore, when assessing for all adaptive and innate immune cell densities, we observed that IMs have strong correlations not only with aDCs and M2 macrophages but also with regulatory T-cells (Tregs) and immune checkpoints (Supplementary Fig. 6), strongly implicating the presence of an immune suppressive environment. We thus define this high CD14 expressing cohort as the 'IM-rich subset' of LUSC.

**TNFα-dependent NFκB activation leads to CCL2-driven IM recruitment**. Given our findings that IMs may promote LUSC metastasis, we sought to functionally characterize IMs in an immune-competent tumor model. To date, the field has lacked an immune-competent mouse model of LUSC that faithfully metastasizes. To address this limitation, we began by characterizing the metastatic properties of the murine LUSC cell line (KLN205) derived from bronchial carcinogen exposure[29] by performing orthotopic injections in syngeneic DBA2 mice. The resulting tumors were poorly differentiated, exhibited central necrosis, and displayed classic IHC patterns of human LUSC (Supplementary Fig. 7a, b). Following several rounds of an in vivo passage selection technique[30,31], we developed sub-clones (LN2-2 and LN4K1) with distinct metastatic properties. Both sub-clones had significantly increased number and frequency of lymph node metastases; however, the LN4K1 sub-clone developed more distant metastases, while LN2-2-injected mice rapidly died from malignant pleural effusions (Supplementary Fig. 7c–g). Although KLN205 and LN4K1 had similar intrinsic growth rates in vitro, LN4K1 tumors grew significantly faster in vivo (Supplementary

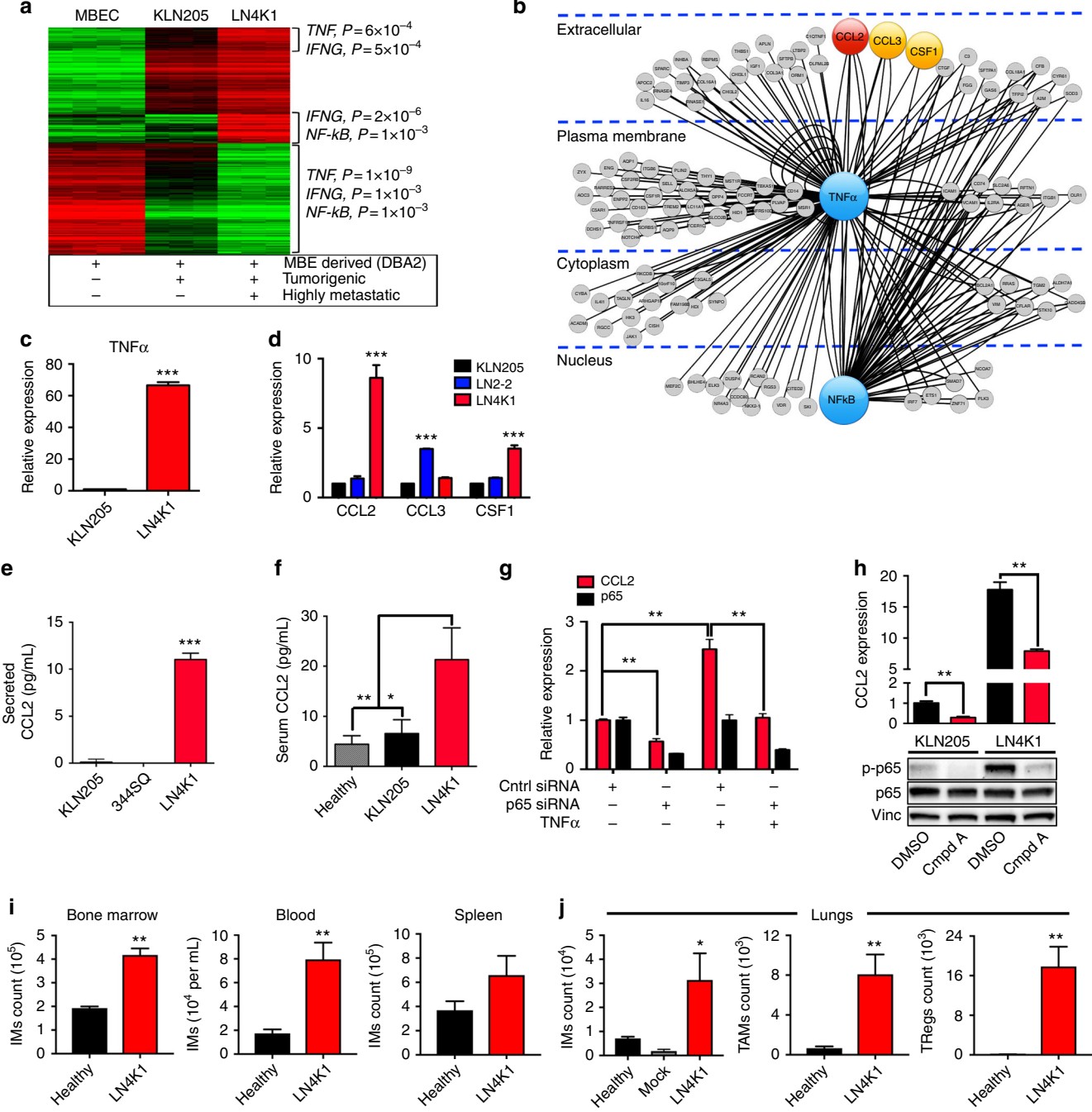

**Fig. 4** TNFα activation of NFκB promotes CCL2-mediated IM recruitment. **a** Microarray expression data (left) comparing murine bronchial epithelial cells (MBECs), parental KLN205 and the LN4K1 sub-clone. Top upstream pathways (right) from Ingenuity Pathway Analysis (IPA) are shown for the differentially regulated genes shown in brackets. **b** An upstream network visualization from IPA of all over-expressed genes (all nodes) in the upper portion of the heat map shown in Fig. 1c with significant log-rank (survival analysis) *p*-value ( < 0.05). TNFα and NFκB (blue nodes) were amongst the top upstream regulators known to have direct roles (black lines) in promoting CCL2, CCL3 and CSF1 chemokines (the degree of redness corresponds with increasing statistical significance). **c** Relative expression of TNFα. **d** CCL2, CCL3 and CSF1 by qPCR. Data are averages ± s.e.m. *P*-values were obtained with Student's t-test in comparison with KLN205. **e** Relative levels of CCL2 as measured by ELISA from secreted media of cells growing in vitro or, **f**, from plasma of tumor-bearing mice. Data are averages ± s.e.m. **g** Relative mRNA expression of CCL2 and p65 in LN4K1 cells following treatment with control or p65 siRNA with or without exogenous TNFα (100 ng/mL). **h** Relative expression of CCL2 mRNA (top) and phospho-p65 and p65 protein (bottom) following treatment with DMSO or an IKKβ inhibitor (Compound A, 5 μM) for 5 h. **i** Relative IM counts in the bone marrow, blood, spleen from healthy DBA2 mice versus those with LN4K1 tumors. **j** Relative IM, TAM and TReg counts from the lungs of healthy versus LN4K1-bearing mice. IMs were also assessed in age-matched DBA2 mice following HBSS 'Mock' injection. *P*-values obtained with one-sided Student's t-test, $n = 5$ mice/group for **f**, **i**, and **j**. * $P < 0.05$, ** $P \leq 0.01$, *** $P \leq 0.001$

Fig. 8a, b), suggesting important differences in the TME. While no changes were observed in angiogenesis, LN4K1 tumors had increased proliferative indices (Supplementary Fig. 8c, d). Similarly, human Secretory tumors exhibited increased proliferative indices relative to Classical tumors, while no significant

differences in angiogenesis were found among the four subtypes of LUSC (Supplementary Fig. 8e, f).

To explore the molecular mechanisms underlying the metastatic properties of LN4K1, we compared the expression profiles of 3 cell lines: murine bronchial epithelial cells (MBEC) isolated

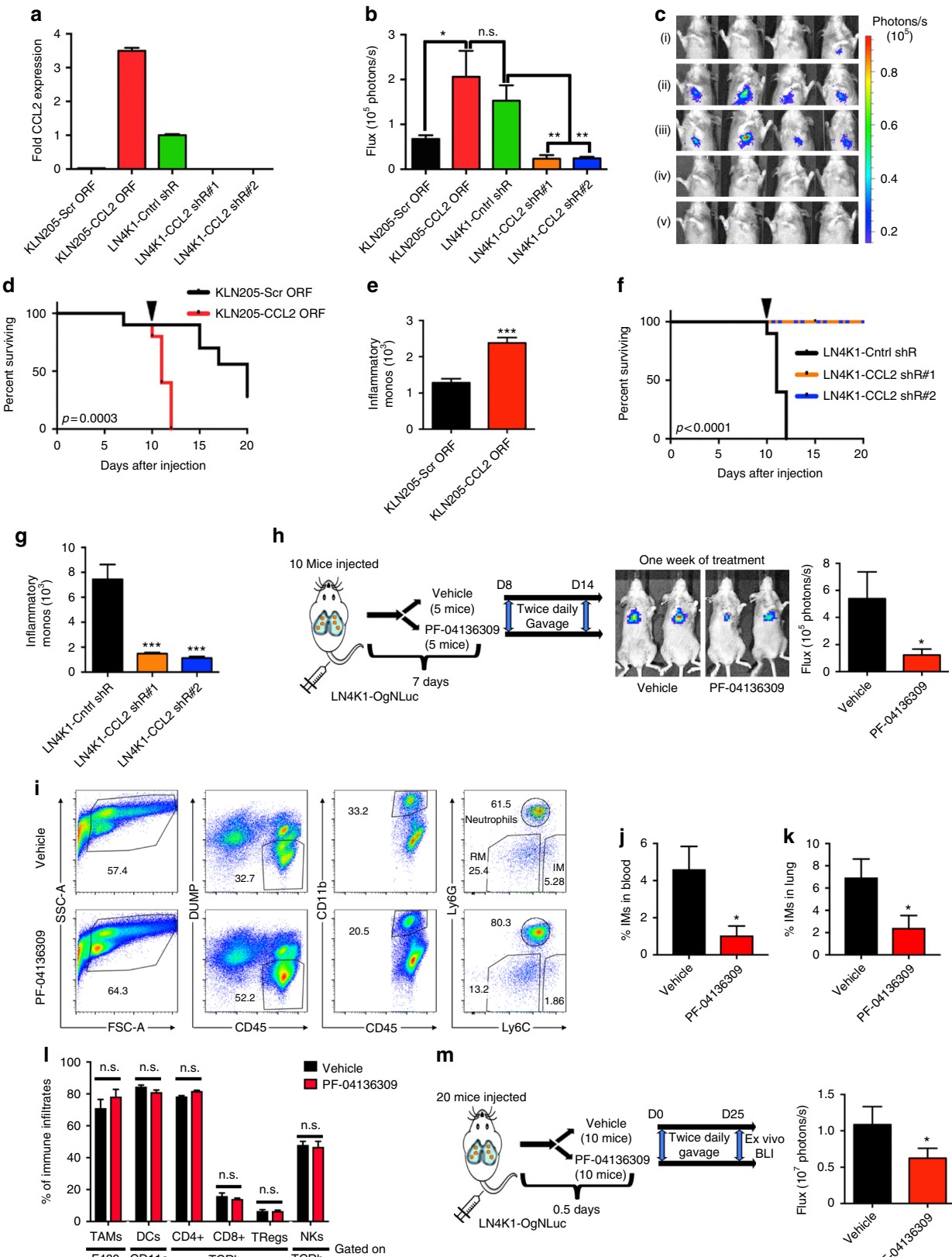

from healthy adult DBA2 mice, KLN205, and LN4K1. Several patterns emerged, the most significant of which were signatures of TNFα, IFNγ, and NFκB signaling activation (Fig. 4a, Supplementary Data 8). Intriguingly, this result matched an IPA screening of the TCGA "survival genes", which also revealed that TNFα ($P < 10^{-34}$) and NFκB ($P < 10^{-11}$) were highly significant and interconnected upstream drivers of the IM chemokines (Fig. 4b, Supplementary Data 9). Higher TNFα expression determined poorer overall survival and was significantly observed in the Secretory LUSC subtype. TNFα expression also corresponded with the pattern of IM chemokines (Fig. 2e, f, Supplementary Fig. 9). Using an analysis comparing our lung squamous carcinoma model and TCGA LUSC RNA-seq data based on the approach used by Xu et al.[32], we observed that i) the two main biological processes (with the most genes and lowest p-values) among genes up-regulated in Secretory as well as in LN4K1 are 'signal transduction' ($P = 2.6 \times 10^{-5}$, 23 genes) and 'inflammatory response' ($P = 4.2 \times 10^{-4}$, 11 genes); and ii) shared inflammation-related genes included CCL2 and TNFα (Supplementary Fig. 10, Supplementary Data 10). Notably, the inflammatory response was also one of the key GO hits for the broader set of genes that are more highly expressed in Secretory (Supplementary Data 4). These results suggest that inflammation-related genes may be triggered, at least partially, by a cell-intrinsic manner in LUSC.

Consistent with these findings, compared with KLN205, LN4K1 had markedly elevated TNFα and CCL2 expression levels and modest increases in CSF1 (Fig. 4c, d). Compared with KLN205 and a metastatic LUAD line (344SQ), LN4K1 secreted abundant levels of CCL2 in vitro (Fig. 4e), which was confirmed in the plasma of tumor-bearing mice (Fig. 4f). Our group previously showed that TNFα-mediated activation of the canonical NFκB pathway directly promotes CCL2 expression[33]. Indeed, silencing p65 significantly decreased basal and exogenous TNFα-mediated stimulation of CCL2 in LN4K1 cells (Fig. 4g). Moreover, compared with KLN205, LN4K1 displayed increased activation of NFκB, and pharmacological inhibition of IKKβ using Compound A (Cmpd A)[34] significantly reduced the levels of phospho-p65 and CCL2 (Fig. 4h).

Next, to characterize the immunologic changes elicited by the LN4K1 model, we performed flow cytometry on different immune populations of mice injected with this cell line versus healthy, non-tumor bearing mice. The LN4K1 model promoted substantial increases in IM generation in the bone marrow (BM), leading to a significant increase in IMs in the blood and a non-significant increase in the spleen (Fig. 4i). The lung TME was characterized by significant increases in IMs, TAMs, and Tregs (Fig. 4j), as well as neutrophils, natural killer cells, and conventional CD4 and CD8 cells and a non-significant increase in DCs.

**CCL2 is necessary and sufficient for enhanced LUSC metastasis.** Considering that metastasis accounts for approximately 90% of cancer-related deaths, there is a surprising paucity of scientific knowledge concerning the specific mechanisms that drive LUSC metastasis. Arguably one of the least understood steps in the metastatic cascade occurs after cancer cells successfully intravasate into the circulation. It is now well recognized that distant colonization is an extremely inefficient process, and most cancer cells that reach distant tissues rapidly undergo apoptosis[35].

Consistent with increased secretion of the CCL2 chemokine (Fig. 4f), compared with the parental KLN205 cell line, LN4K1 was associated with marked increases in IM recruitment and rapid development of distant metastases (Supplementary Fig. 11). Given the robust immunologic response that LN4K1 invokes in the lung TME, we addressed whether the CCL2-mediated recruitment of IMs is necessary and sufficient for distant metastasis development in LUSC using an experimental metastasis model. Indeed, the stable overexpression of CCL2 in KLN205 was sufficient to account for the enhanced metastatic properties of LN4K1, while the silencing of CCL2 in LN4K1 with two different shRNAs had the opposite effect, substantially decreasing metastatic properties (Fig. 5a–c). Additionally, consistent with the effects of CCL2 on metastasis and IM recruitment, CCL2 overexpression in KLN205 led to significantly shorter survival and increased IM infiltration in the lungs (Fig. 5d, e). Conversely, silencing CCL2 in LN4K1 dramatically extended survival, which corresponded with significantly decreased IM infiltration (Fig. 5f, g). To assess how robust CCL2-mediated recruitment of IMs is on LUSC progression, we developed an additional model of LUSC metastasis. Using the parental KAL cell line, which was derived from a kinase-dead IKKα genetically-engineered mouse (GEM) model of LUSC[36], we performed two rounds of in vivo selection as described for the LN4K1 model (Supplementary Fig. 7c). With this approach, we developed the KAL-LN2E1 metastatic sub-clone, which forms large, poorly differentiated orthotopic LUSC tumors and rapidly develops lymph node and chest wall metastases (Supplementary Fig. 12a, b). To corroborate our findings with the LN4K1 model, we assessed whether IMs contribute to LUSC metastases independent of their suppressive role on T-cells[37]. We generated stable KAL-LN2E1 lines expressing CCL2 shRNA hairpins and orthotopically injected them into NSG mice (Supplementary Fig. 12c). While no effect was observed on primary tumor development (not shown), both groups injected with shCCL2 KAL-LN2E1 lines showed decreased numbers and incidence of distant metastases, consistent with decreased IM recruitment in the primary tumor (Supplementary Fig. 12d–f).

**Fig. 5** CCL2-mediated IM recruitment is critical for LUSC metastasis. **a** Relative expression of CCL2 for KLN205 and sub-clones. **b** Quantification of luciferase signal. **c** Representative images obtained 10 days after cell injection of (i) KLN205-Scr ORF, (ii) KLN205-CCL2 ORF, (iii) LN4K1-Cntrl shR, (iv) LN4K1-CCL2 shR#1 and (v) LN4K1-CCL2 shR#2. Data are averages ± s.e.m. P-values were obtained with Student's t-test, $n = 10$ mice/group. **d** Survival plots of mice following tail vein injection of KLN205 cell lines. The black arrow indicates tissue harvest, $n = 10$ mice/group. **e** Number of IMs per lung lobe, $n = 12$ lobes/group. **f** Survival plots of mice following tail vein injection of LN4K1 cell lines. The black arrow indicates tissue harvest, $n = 10$ mice/group. **g** Number of IMs per lung lobe, $n = 12$ lobes/group. **h** Schematic (left) and quantification of luciferase signal (right) of mice treated with vehicle or PF-04136309 to assess effects on established metastases. **i** FACS plots and (**j**) quantification of percent IMs in the blood and (**k**) right lung of LN4K1-bearing mice. Data are averages ± s.e.m. P-values were obtained with Student's t-test, $n = 5$ mice/group. **l**, FACS analysis of percent immune infiltrates for TAMs (gated on F480), DCs (gated on SiglecF-/CD11c), CD4, CD8, Tregs (gated on TCRb + ) and NK cells (gated on SiglecF-/B220-/TCRb-). **m** Schematic (left) and quantification of luciferase signal (right) of mice treated at the time of cell injection to assess effects on preventing metastasis. Data are averages ± s.e.m. P-values were obtained with Student's t-test, $n = 10$ mice/group. n.s. = non-significant, * $P \leq 0.05$, *** $P < 0.001$. For panel **b**, * FDR < 0.05, ** FDR < 0.01

**Development of a molecular strategy for targeting LUSC IMs.**
We then assessed the therapeutic efficacy of targeting IMs in our metastatic LUSC model. While anti-CCL2 antibodies have shown initial promise in breast cancer[11], a rebound effect that accelerates metastasis has been observed upon drug cessation and this therapy is no longer being clinically developed[38]. Furthermore, other chemokines such as CSF1 and CCL3 can have redundant properties in recruiting IMs and TAMs[16,24]. Thus, targeting IMs requires a more effective strategy. Recent studies with a CCR2

inhibitor (PF-04136309) have demonstrated effective blockade of IM recruitment in pancreatic cancers[13,39]. To assess the effects of PF-04136309 on established LUSC metastases, one week following tail vein injection of luciferase-labeled LN4K1 cells, mice were treated with vehicle or PF-04136309 and imaged one week later (Fig. 5h). Significant reduction in lung metastasis was observed with PF-04136309 treatment, consistent with significant reductions in both circulating and lung TME IMs (Fig. 5h–k, Supplementary Fig. 13). However, there were no significant changes in

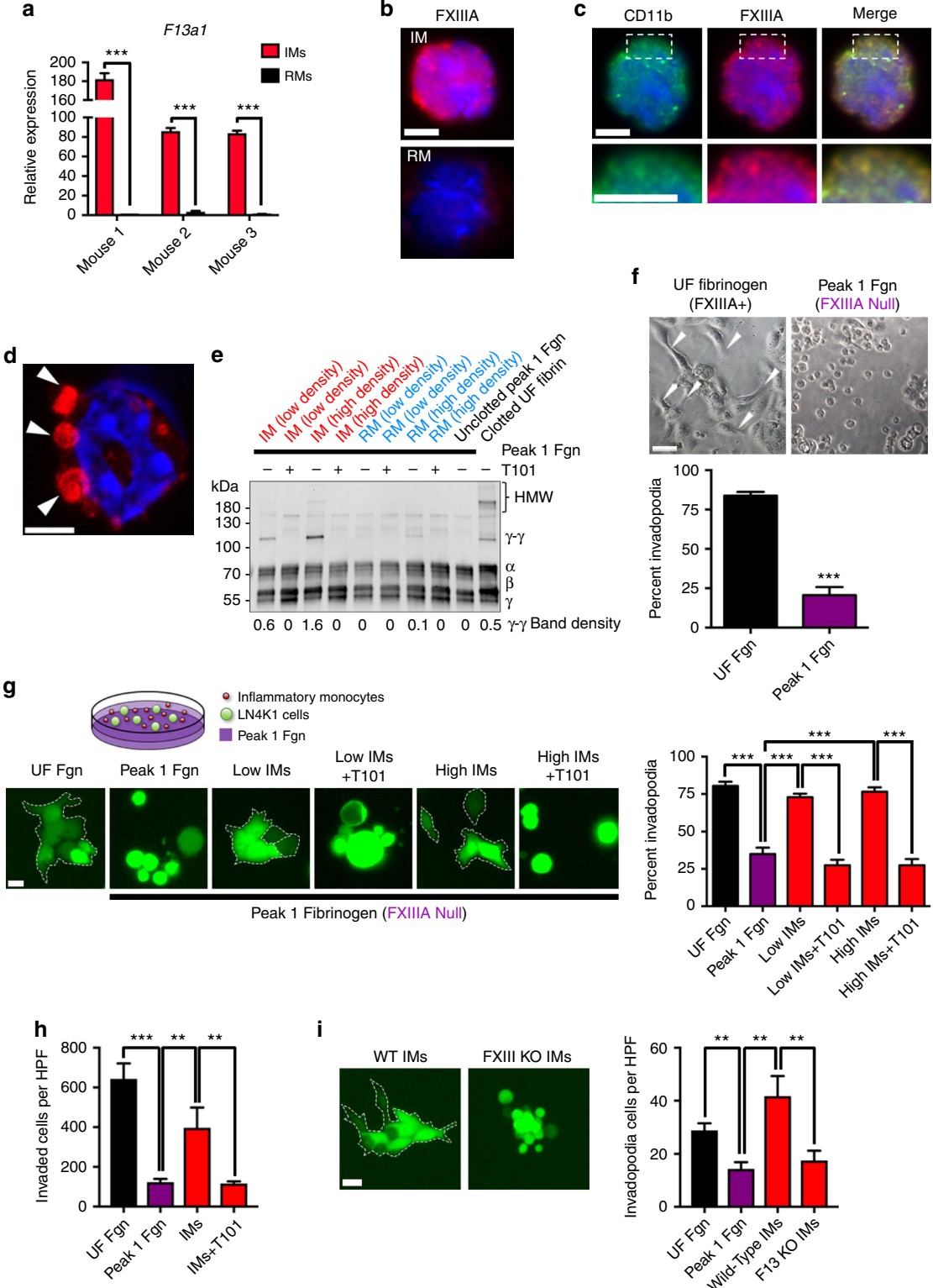

TAMs, DCs, NKs, or Tregs (Fig. 5l). In accordance with a prior study using CCR2 inhibition[13], 1 week of treatment did not significantly affect CD206$^{High}$ or CD206$^{Low}$ TAM subsets (Supplementary Fig. 14a). Additionally, we found no significant effects on recruitment of gamma-delta T-cells (Supplementary Fig. 14b), which express CCR2 and are involved in IFNγ production[40]. These results strongly suggest that the therapeutic effects arose from blocking IM recruitment into the TME. Additionally, CCR2 blockade with PF-04136309 significantly inhibited metastasis development when treatment was initiated on the day of cancer cell injection (Fig. 5m).

**IMs cause FXIIIA-mediated fibrin cross-linking and LUSC progression.** We next sought to investigate the molecular mechanism by which IMs promote LUSC metastasis. Over the past decade there has been rapid growth in our understanding of how perivascular macrophages promote tumor growth and metastasis[18]; however, the contribution of IMs prior to differentiating into macrophages remains poorly understood. Given the recent findings that RMs have opposing, anti-metastatic functions when compared with IMs[17], we hypothesized that divergent expression patterns of IMs and RMs may reveal important differences in their biology. In a previously performed transcriptome profile of IMs and RMs in the blood and spleen[41], we observed that *F13a1* had the sharpest differential expression pattern (expression in IMs > RMs) for both tissue compartments. *F13a1*, which encodes for factor XIII-A subunit (FXIIIA), cross-links fibrin and other substrates and has critical roles in blood clot stabilization and wound healing[42]. Compared with RMs, we confirmed that IMs express markedly increased levels of *F13a1* mRNA (Fig. 6a). Importantly, comparing the expression levels of *F13a1* mRNA in CD206$^{High}$ and CD206$^{Low}$ TAM subsets revealed that TAMs express *F13a1* levels that are similar to RMs (Supplementary Fig. 15). We also found a strong level of positive correlation of expression among *F13a1*, CD14, and our IM-rich subset at the transcriptional level (Supplementary Fig. 16a). Neither VEGFa nor the well-validated myeloid-derived suppressor cell (MDSC) mediators *Arg1* and *NOS2* exhibited such a correlation with CD14 or the IM-rich subset (Supplementary Fig. 16a). Furthermore, compared to tumors with low CD14 expression, *F13a1* expression levels in CD14 high tumors were more than 2.5-fold higher (Supplementary Fig. 16b). In agreement with the Secretory subtype frequently having high CD14 expression (Fig. 2b, Supplementary Fig. 16a), we found that this subtype also associated with above median expression of *F13a1* (Supplementary Fig. 16c). These findings implicate IMs as a rich source of *F13a1* in the tumor microenvironment of LUSC.

Immunofluorescent staining revealed that FXIIIA protein is produced at higher levels in IMs than RMs (Fig. 6b), and FXIIIA localizes with CD11b near the cell surface (Fig. 6c). Using confocal microscopy, we observed dense deposits of FXIIIA near podosome-like structures in IMs (Fig. 6d, Supplementary Video). Using IM and RM cell densities observed in the TME of our LN4K1 model, we found that IMs induced fibrin cross-linking (γ-γ formation) when added to FXIIIA-depleted (Peak 1) fibrin (ogen), and this activity was fully inhibited with a FXIIIA inhibitor, T101 (Fig. 6e). In contrast, only subtle cross-linking was seen with even high RM densities (Fig. 6e).

Previously, others have shown that cancer cells can utilize cross-linked fibrin to form invadopodia[43]. To test the contribution of FXIIIa activity to tumor cell function, we interrogated LN4K1 cell invadopodia formation. We found that LN4K1 cells could easily form invadopodia when grown in unfractionated fibrinogen (which contains FXIIIA); however, this was significantly attenuated when placed in FXIIIA-depleted fibrinogen (Fig. 6f). We then hypothesized that IMs in the TME provide the necessary FXIIIA(a) activity to cross-link fibrin and create a scaffold for cancer cell invasion. Using a co-culture model of GFP-labeled LN4K1 cells and freshly isolated IMs, we observed that both the low and high densities of IMs could rescue invadopodia formation in FXIIIA-depleted fibrinogen to a degree similar to that of unfractionated fibrinogen (Fig. 6g). T101 treatment completely abolished this effect, implicating FXIIIa activity in this mechanism (Fig. 6g). Next, we determined whether the increased cancer cell invadopodia formation induced by FXIIIA-expressing IMs also corresponded with increased LUSC invasion. Using trans-well invasion chambers, we observed a significant reduction in the invasive capabilities of LN4K1 cells when grown in FXIIIA-depleted fibrinogen compared with unfractionated fibrinogen (Fig. 6h). Similar to the invadopodia assays, this phenotype in FXIIIA-depleted fibrinogen was significantly rescued in the presence of IMs and abolished in the presence of T101 (Fig. 6h). Importantly, the effects of T101 on invasion were not seen when performed in Matrigel (Supplementary Fig. 17), suggesting the importance of the fibrin cross-linking context. Finally, to assess the importance of FXIIIA expression in IMs for promoting cancer cell invasion, we performed the invadopodia assay using IMs from age-matched wild-type or *F13a1* knock-out mice. Consistent with our prior findings, the presence of wild-type IMs led to substantial increases in LN4K1 cells with invadopodia formation, while co-culture with *F13a1*$^{-/-}$ IMs completely abolished this phenotype (Fig. 6i).

To assess FXIIIA activity within the LUSC TME and its association with disease progression and metastasis, we developed

**Fig. 6** Factor XIIIA in IMs promotes fibrin cross-linking and LUSC invasion. **a** Relative mRNA expression of *F13a1* from sorted IMs and RMs, $n = 3$ mice. **b** Immunofluorescent (IF) imaging of IMs and RMs comparing FXIIIA (red) expression. **c** Dual staining for FXIIIA (red) and CD11b (green) in IMs. Contents of dotted white box are enlarged under each panel. **d** Confocal imaging of IMs for FXIIIA (red). White arrows point toward podosome-like structures. Scale bar: 5 μm; nuclei were stained with Hoechst (panels **b**–**d**). **e** Fibrin cross-linking patterns by western blotting using a polyclonal anti-human fibrin(ogen) antibody using freshly sorted IMs and RMs (Low = 25k cells, High = 100k cells), with or without the FXIIIA-inhibitor, T101. **f** Percent invadopodia of LN4K1 cells growing in either unfractionated (UF Fgn, left) or Peak 1 (FXIIIA-depleted) fibrinogen (right). White arrows show evidence of invadopodia. Scale bar 50 μm. Data are averages ± s.e.m. *P*-values obtained with Student's t-test. **g** Schematic of co-culture model (top). Representative images of LN4K1-GFP cells growing in either UF Fgn or Peak 1 with or without low (100k) or high (300k) IMs. FDR = 0.0001 for all statistical comparisons shown. Groups in **e** + **g** treated with the T101 were dosed at 50 μM. Scale bar 12.5 μm. Data are averages ± s.e.m. *P*-values were obtained with Student's t-test. **h** Invaded LN4K1-GFP cells per high power field (HPF) at 24 h following seeding into either UF Fgn or Peak 1 Fgn alone or co-cultured with IMs with or without T101 (50 μM). FDR < 0.01 for all statistical comparisons shown. **i** Representative images (left) of LN4K1-GFP cells co-cultured with IMs from either wild-type or *F13a1*$^{-/-}$ mice. Number of invadopodia positive LN4K1-GFP cells (right) per HPF at 24 h when growing in UF Fgn or Peak 1 Fgn alone or co-cultured with IMs from either wild-type or *F13a1*$^{-/-}$ mice. Scale bar 12.5 μm. Data are averages ± s.e.m. *P*-values were obtained with Student's t-test. FDR < 0.01 for all statistical comparisons shown. ** *P* < 0.01, *** *P* < 0.0001

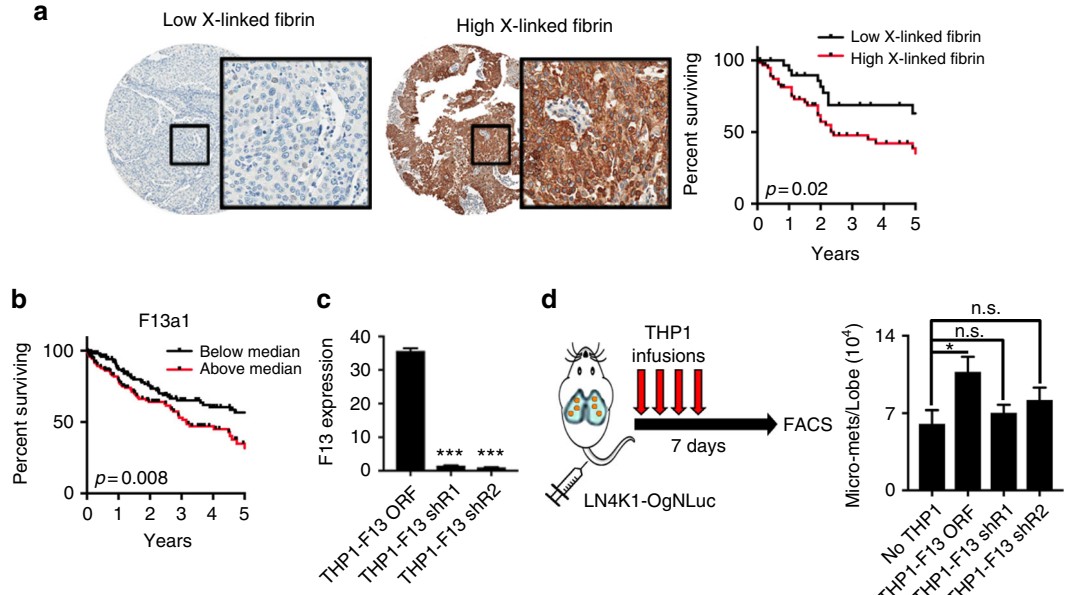

**Fig. 7** FXIIIA and fibrin cross-linking is associated with LUSC progression. **a** Representative IHC images (left) of tumors from the LUSC tissue microarray with low or high cross-linked fibrin (n = 96 patients). Kaplan-Meier plot of recurrent-free survival following surgical resection. P-values were obtained with the log-rank test. **b** Kaplan–Meier plot of overall survival in LUSC patients split by median expression levels of F13a1. P-values were obtained with the log-rank test. **c** Relative expression levels of F13a1 in THP1 monocytes stably expressing either F13 ORF or shRNAs against F13a1. P-values were obtained with Student's t-test. **d** Schematic (left) and quantification of distant micro-metastases (right) following infusions of THP1 monocytes. NSG mice were injected via tail vein with LN4K1-OgNLuc cells on Day 1, followed by daily infusions of THP1 cells in the respective groups for Days 1–4. Mice were necropsied on Day 7, lungs were dissociated and FACS analysis performed for EpCAM + cells to quantify micro-metastases per lobe. Data are averages ± s.e.m. P-values were obtained with Student's t-test, n = 3 mice/group. n.s. = non-significant, * P < 0.05, *** P < 0.001

an IHC protocol with a novel monoclonal antibody that specifically detects the cross-linked fibrin neo-epitope (Supplementary Fig. 18a). Using a TMA of 96 surgically-resected LUSC tumors, we found that compared with low or intermediate staining, high staining of intra-tumoral fibrin cross-linking was associated with significantly worse recurrence-free survival (Fig. 7a, Supplementary Fig. 18b). Consistent with this observation, LUSC tumors from the TCGA dataset expressing high *F13a1* had significantly worse survival (Fig. 7b). Finally, to assess whether FXIIIA over-expressing monocytes are sufficient to enhance LUSC metastasis in vivo, we stably over-expressed or silenced FXIIIA in a THP1 monocyte model (Fig. 7c). Genetically modified THP1 monocytes were infused into NSG mice daily for a total of 4 days following intravenous injection of LN4K1, and at 1 week the lungs were dissociated and micro-metastases were enumerated using FACS for EpCAM. Compared with LN4K1 alone injected mice, the only group with significantly increased metastases were the mice treated with THP1-F13 ORFs, while neither F13 shRNA-expressing groups showed an increase in metastases (Fig. 7d).

Taken together, we have uncovered a previously unappreciated 'IM-rich subset' of LUSC that is driven by a TNFα-NFκB-CCL2 axis of IM recruitment. These IMs express high levels of FXIIIA, which facilitates LUSC cell invasion and disease progression by promoting fibrin cross-linking (Fig. 8).

## Discussion

PD1/PD-L1 immune checkpoint inhibitors, while effective in several cancer types, provide only about 20% response rates in unselected LUSC patients[4,5]. Thus, there is an urgent need to extensively characterize other immunologic mediators of LUSC progression, which may unveil logical, non-overlapping combination approaches to treat this disease.

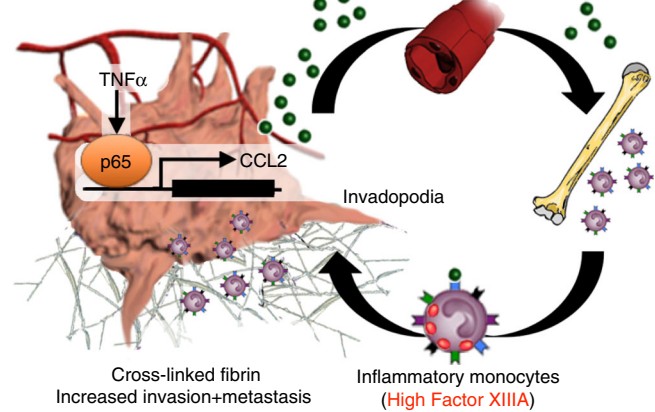

**Fig. 8** Schematic of the 'IM-rich subset' of lung squamous carcinoma. TNFα activation of the canonical NFκB leads to LUSC cell secretion of chemo-attractant CCL2, which stimulates the bone marrow to release inflammatory monocytes (IMs) into circulation. The IMs bring large payloads of FXIIIA into the tumor microenvironment, leading to cross-linked fibrin, LUSC invadopodia formation and progression

By integrating multi-step genomics analyses with novel mouse models, we found that IMs are critical drivers of LUSC metastasis. Our computational analyses aimed to (1) determine the specificity of gene expression in patient subsets, (2) narrow down the pool of genes that are involved according to biological and clinical relevance, and (3) use information available in biological databases about gene-gene interactions, experimentally characterized gene sets, and gene biological functions to gain insight into LUSC. This analysis clearly implicated CD14 and the IM-rich subset in the poor survival of LUSC patients. Remarkably, the significance of IMs on overall patient survival was greater than that of TAMs

and aDCs. This finding was surprising given that IMs are traditionally thought of as inactive precursors that develop into macrophages, DCs, and MDSCs in response to external cues. TAMs have been studied in many syngeneic cancer models and have been associated with the promotion of metastasis[16,18,44]. However, how IMs mechanistically contribute to the establishment of metastasis remains poorly understood. Reasonable hypotheses are that the CCL2-elicited IMs may play similar roles to TAMs by providing both a pro-tumor, pro-metastatic microenvironment through the secretion of growth factors as well as by being highly immunosuppressive, thus preventing an anti-tumor immune response. The surface flow markers used to identify IMs in this and other studies may also identify monocytic myeloid-derived suppressor cells (m-MDSCs), a subset of bone marrow elicited myelo-monocytic cells that are closely related if not identical to IMs; the field is still debating if there are subtle differences in gene expression or suppressive capabilities between IMs and m-MDSCs[11,24,38,45]. When studied functionally, tumor-localized m-MDSCs are highly immunosuppressive, particularly against cytotoxic T-cells, largely through their secretion of iNOS and Arginase 1[46]. Along these lines, a recent review discussing MDSC nomenclature and characterization suggests that IMs can be attributed to m-MDSCs[37]. However, our results suggest that IMs actively participate in the LUSC metastatic process and may have unique functions in the TME independent of this immunosuppressive phenotype. In support of this possibility, we found no significant association between iNOS or Arginase 1 and our 'IM-rich subset' of LUSC tumors.

Utilizing immunogenomic approaches, we found that the 'IM-rich subset' can be used to re-classify the four LUSC subtypes using median CD14 expression as the threshold. To further evaluate these findings, we developed a novel immune-competent metastasis model of LUSC. This model is characterized by IMs recruited by LUSC tumor cells through TNFα-mediated activation of NFκB signaling, which promotes the secretion of the monocyte chemokine CCL2. Integrated network analyses of genes linked to worse survival for LUSC also revealed enrichment for activation of a TNFα-NFκB-CCL2 signaling axis. Although TNFα within the TME is often derived from cellular constituents of the TME itself[47], we observed marked elevations of TNFα in our metastatic LN4K1 sub-clone developed through serial in vivo passages. This finding suggested that metastatic LUSC sub-clones may secrete elevated levels of TNFα, and at least during the initial stages of metastasis, LUSC cells may play an autocrine role in the heightened secretion of CCL2 with subsequent recruitment of IMs. Because IMs and IM-derived TAMs also secrete copious amounts of TNFα, our data imply a feed-forward loop whereby LUSC tumors promote their own secretion of CCL2 followed by the recruitment of TNFα-secreting IMs.

After determining that CCL2 is necessary and sufficient to promote distant LUSC metastasis, we evaluated a therapeutic strategy for targeting IMs. Specifically, we targeted the main CCL2 cell surface receptor, CCR2, using a potent, clinic-ready CCR2 inhibitor (PF-0436309)[13,39]. This strategy significantly reduced both the blood and tumor levels of IMs, which inhibited the seeding and initial growth of LN4K1 metastasis and prevented the progression of established LUSC metastases. This approach unambiguously shows the potential of targeting this immune cell type for the treatment of LUSC patients.

Our study of an "IM-rich subset" of LUSC is reminiscent of the paradigm that tumors resemble "wounds that do not heal"[48]. The main tenets of how tumor-induced "wounds" form are as follows: (1) VEGFa promotes angiogenesis and increased vascular permeability, (2) this in turn leads to extravasation of fibrinogen and several classes of lymphocytes, (3) activation of the coagulation cascade occurs, (4) fibrin deposition takes place, and (5) an

irregular collagen matrix forms. Although this process is analogous to physiologic wound healing, tumor-induced "wounding" is usually irreversible and leveraged by cancer cells to effectively parasitize the host organ[49]. Along these lines, we found that IMs express high amounts of Factor XIIIA, which rapidly and potently leads to fibrin cross-linking in the TME, and this evidence points to a novel and important mechanism of LUSC progression. Consistent with these findings, plasma levels of fibrin degradation products (D-dimers) have been linked with higher stage lung cancers and markedly worse prognosis[50,51]. Moreover, plasma FXIIIA has been shown to promote cancer by inhibiting the intravascular clearance of natural killer cells[52]. Importantly, none of these prior studies have suggested a role for cellular FXIIIA in IMs for tumor progression. In agreement with observations that FXIIIA has pleiotropic roles in mediating wound healing[42], our findings point to a previously unappreciated mechanism by which tumors represent "non-healing wounds".

IM-derived FXIIIA-mediated fibrin cross-linking creates an important scaffold for cancer cell invasion. Targeting this IM population in an immunocompetent LUSC model had substantial effects on blocking metastases. Moreover, dense intra-tumor deposits of cross-linked fibrin in resected LUSC tumors were associated with poor survival. Given the rapidly evolving landscape of precision immune-oncology, these findings identify IMs as a novel context-specific vulnerability of LUSC and provide an important insight into the mechanisms through which this immune cell type determines a poor prognosis.

## Methods

**Cell lines and maintenance**. All cell lines were maintained in 5% $CO_2$/95% air at 37 °C. KLN205 lung squamous cell carcinoma cells were obtained from the ATCC, parental KAL cells were kindly provided by Dr. Yinling Hu (National Cancer Institute, Frederick, MD) and 344SQ lung adenocarcinoma cells were kindly provided by Dr. John Kurie (M.D. Anderson Cancer Center; Houston, TX). THP1 monocytes were kindly provided by Dr. Gianpietro Dotti (University of North Carolina, NC). KLN205 cells and derived sub-clones (LN2-2 and LN4K1) were maintained in MEM and 344SQ cells were maintained in RPMI 1640, both supplemented with 10% fetal bovine serum (FBS) and 1% Penicillin Streptomycin. KAL cells and their derivatives were maintained in DMEM supplemented with 10% FBS. Mouse bronchial epithelial cells (MBECs) were isolated from three healthy adult DBA2 mice. All cell lines were tested to confirm the absence of *Mycoplasma*, and all in vitro experiments were conducted with 60–80% confluent cultures.

**Establishment of sub-clone cell lines**. Metastatic lesions were mechanically minced in RPMI 1640 containing 0.125% collagenase III and 0.1% hyaluronidase under laminar flow within a hood and using a sterile scalpel. Cells were then pelleted, resuspended in 0.25% trypsin for 20 min at 37 °C with vortexing every 5–7 min, and placed in a T75 flask with complete MEM medium.

**Lentivirus Packaging and Infection**. Lentiviral particles for CCL2 overexpression in KLN205 cells were purchased from GeneCopoeia: Scr ORF (pReceiver-Lv152 Negative Control Lentifect Purifed LV Particles) and CCL2 ORF (CCL2 (NM_011333.3) Lentifect Purifed LV Particles). Lentiviral vectors for CCL2 knockdown in LN4K1 and KAL-LN2E1 cells were also purchased from GeneCopoeia: Cntrl shR (CSHCTR001-1-LVRU6H), CCL2 shR#1 (MSH030124-1-LVRU6H), CCL2 shR#2 (MSH030124-2-LVRU6H), CCL2 shR#3 (MSH030124-3-LVRU6H), and CCL2 shR#4 (MSH030124-4-LVRU6H). Lentiviral particles for Factor 13 overexpression and silencing in THP1 cells were purchased from GeneCopoeia: Factor 13 ORF (NM_028784.3), Factor 13 shR#1 (HSH005069-1-LVRU6H) and Factor 13 shR#2 (HSH005069-2-LVRU6H). OgNLuc vector was a kind gift from Dr. Antonio Amelio (Lineberger Comprehensive Cancer Center; UNC Chapel Hill, NC). Lentivirus was produced by transfecting human embryonic kidney cells (293 T) with lentiviral vector, packaging plasmid (psPAX2) and envelope plasmid (pMD.2G). Media was changed the next day, and 2 days later viral supernatant was collected and filtered to remove cellular debris. Cells were infected with lentiviral particles overnight using Polybrene and were then selected with growth medium containing 200 μg/mL hygromycin (for shR and ORF lentiviruses for each respective cell line) and 1 μg/mL puromycin (for OgNLuc lentivirus).

**Animals, in vivo models and tissue processing**. Adult DBA/2, NSG and FVBn mice were purchased from Jackson Labs. Factor XIIIa knock-out and wild-type C57Bl/6 mice were obtained from Dr. Alisa Wolberg. These animals were cared for

according to guidelines set forth by the American Association for Accreditation of Laboratory Animal Care and the U.S. Public Health Service policy on Human Care and Use of Laboratory Animals. All mouse studies were approved and supervised by the University of North Carolina at Chapel Hill Institutional Animal Care and Use Committee. All animals used were between 6-10 weeks of age at the time of injection. For all animal experiments, cells were trypsinized, washed and resuspended in Hanks balanced salt solution (HBSS; Gibco) prior to injection. Lung squamous KLN205, LN4K1 and KAL-LN2E1 cancer cells were injected either subcutaneously over the posterior flank (KLN205 and LN4K1: $1 \times 10^5$ cells in 100 μL HBSS), intravenously (LN4K1: $1 \times 10^5$ or $3 \times 10^5$ cells in 100 μL HBSS based on experiment) or by an intra-pulmonary technique [KLN205 and LN4K1: $2.5 \times 10^4$ cells and KAL-LN2E1: $5 \times 10^5$, in 100 μL 1:1 mixture of HBSS and BD Matrigel (BD Biosciences)]. For the intra-pulmonary injections, mice were anesthetized with ketamine (80 mg/kg) + xylazine (8 mg/kg) + acepromazine (1 mg/kg) and placed in the right lateral recumbency. Following fur removal and sterile skin preparation, an incision parallel to the rib cage between ribs 10–11 was made to visualize the lung through the intact thoracic pleura. A 1 mL tuberculin syringe with a 30-g needle was used to inject the cell suspension directly into the lung parenchyma at the left lateral dorsal axillary line. After injection, the skin incision was closed using surgery clips and the mice were turned on the left lateral recumbency and observed until fully recovered. Caliper measurements of subcutaneous tumor growth were taken twice weekly and tumor volume was calculated as L X W$^2$ where L is the greatest cross-sectional length across the tumor and W is the length perpendicular to L. Luciferase-labeled tumor progression was monitored once or twice weekly using an IVIS Lumina optical imaging system and Nano-Glo Luciferase Assay substrate (Promega) as per the manufacturer's instructions. For CCR2 inhibitor (PF-04136309, Pfizer) experiments, the drug was prepared fresh every 3–4 days and administered at a dose of 200 mg/kg/mouse, twice daily by oral gavage. For adoptive infusion of THP1 cells, NSG mice were purchased from Jackson Labs. LN4K1 cells were infused ($1 \times 10^5$ cells in 100 μL HBSS) on day 1, and THP1 monocytes expressing either Factor 13 ORF or Factor 13 shRNAs were infused ($1 \times 10^6$ cells in 100 μL HBSS) in their respective groups on days 1-4. On day 7, mice were sacrificed and the lung lobes were dissociated mechanically and enzymatically into single cell suspensions. Micro-metastasis enumeration was performed by staining cancer cells with anti-mouse EpCAM-APC antibody (ab95641) and performing FACS analysis. In all experiments 3–10 mice per group were used. Once mice in any group became moribund, they were all sacrificed, necropsied, and tumors were harvested. Tumor weights, number and location of lymphatic and distant metastases were recorded. Lungs were insufflated with 10% neutral buffered formalin. Tissues used for immunohistochemistry analysis were fixed in 10% neutral buffered formalin, and embedded in paraffin.

**Collection of gene expression and survival data**. Numerical and clinical data of lung squamous cancer (LUSC) samples were downloaded from TCGA Data Portal (https://tcga-data.nci.nih.gov/docs/publications/tcga/) and the Firehose Broad GDAC (Genome Data Analysis Centers) data hub (https://gdac.broadinstitute.org/) and checked for mutual consistency. For all the analyses based on RNA-Seq data, the only samples used were those having numerical data available (on January 2014) as for messenger RNA (mRNA), human methylation (HuMet), copy number variation (CNV) and micro-RNAs (miRNAs) (348 total), in order to exclude samples not sufficiently characterized by TCGA. For maximizing the statistical power of our subtype-based (see section "Assignment of tumor subtypes") survival analyses of LUSC patients (Fig. 1), we broadened this number to include all samples (380 total) (i) for which clinical information had been published by TCGA, (ii) with available mRNA sequencing (see section "Assignment of tumor subtypes") and (iii) having usable survival data (see the next section).

**Survival data pre-processing**. We extracted from the clinical annotation files of LUSC samples (downloaded on January 2014) four types of survival data: (1) "Days to death", (2) "Days to last follow-up", (3) "Days to last known alive", and (4) "Vital status". The processing was the following: (i) disregarding patients (5 total) having a negative value for their "Days to last follow-up" (since negative values are incompatible with survival analyses and this data conflict could not be solved differently); (ii) attributing a vital status of 1 to patients recorded as "dead" and of 0 to patients recorded as "alive"; (iii) using as "Survival days" for "dead" patients the "Days to death"; (iv) using as "Survival days" for "alive" patients the maximum between "Days to last follow-up" and "Days to last known alive"; (v) transforming the survival days into years by using the conversion formula years = days/365. For two patients that, at the time of this analysis, had a positive value of their "Days to death" (i.e., a value different from NA (Not Available), which should be always associated with "alive" patients), we solved the conflict looking at TCGA clinical data (biotab type); after this check, the two patients were both deemed to be "alive".

**Assignment of tumor subtypes**. Normalized RSEM (abbreviation for RNA-Seq by Expectation Maximization) gene expression values of lung squamous cell carcinomas ($n = 491$ tumors) were obtained from TCGA[7,53] (data available on March 2014). Cohort expression values were log2 transformed and median centered by gene. Expression subtypes were predicted in this cohort utilizing a published lung squamous cell carcinoma expression subtype predictor[8], which is very consistent

across distinct genomic platforms (Supplementary Fig. 1). Specifically, the tumor expression data were reduced to genes in common with the predictor centroids. Pearson correlations were calculated between each tumor and the predictor centroids of the four squamous subtypes (Classical, Basal, Primitive, Secretory). For every sample, the subtype assignment was defined as a tumor's largest correlation value (Supplementary Data 11).

**Analysis of RNA-Seq data: definition of a spectrum of gene expression among the subtypes**. We adopted the normalization method that was chosen by TCGA Consortium for RNA-Seq data (RNA-Seq V2 pipeline), based on data quantified using RSEM[53]. Genes listed as such in these files are 20,531 per sample (level 3 of the RNA-Seq data published by TCGA Data Portal). Then, subtypes were aggregated based on two criteria: (i) correlation among the expression values of their samples and (ii) number of samples available. Specifically, the correlation was evaluated as follows: (a) the expression values of all genes of this platform are averaged inside each subtype. At the end of this procedure, the vectors obtained are $V_B$, $V_C$, $V_P$, and $V_S$, for the subtypes Basal, Classical, Primitive and Secretory, respectively, each having 20,531 elements; (b) Pearson's correlation coefficients are calculated among all possible (i.e., six) couples of these four vectors; (c) the two highest and mutually exclusive Pearson's correlation coefficients are used for aggregating these subtypes: at this stage, it is found that the Basal subtype has its closest similarity with the Classical subtype and the same happens between the Primitive and Secretory subtypes. Then, the information about the number of samples per subtype was used for deciding in which order they would be displayed in each heat map (HM), unless differently specified, so that the two subtypes with most samples are plotted at the two HM sides and the other two are displayed in the middle: at this stage, it is defined that the first potential aggregate of subtypes is Classical-Basal (CB, left side of the HM) and the second is Primitive-Secretory (PS, right side of the HM).

**Analysis of RNA-Seq data: identification of a set of representative genes**. In this hybrid analysis, genes were considered (i) differentially expressed between CB and PS, (ii) heterogeneously expressed across the available samples and (iii) qualitatively satisfactory with respect to: (a) their median has an appreciable change between CB and PS samples. Specifically, after determining which, between CB and PS samples, has the largest and smallest median, the relative variation between CB and PS is calculated with respect to the smallest between these two values (precisely, (maximum-minimum)/(minimum + MATLAB epsilon (eps function))). This ratio expresses the relative variation of the median for a gene and the same gene is selected when this variation strictly belongs to the top 50% across all genes; (b) they have a "calculable" p-value according to the MATLAB implementation of the Wilcoxon rank sum test (i.e., a p-value different from 'Not a Number' (NaN), which is the output of the MATLAB function performing this statistical test when all values of the first and second group are identical) < 0.01 and, at the same time, whose p-value is considered 'relevant' for multiple hypothesis testing (the only genes included are those having a number of values ≥ 0.5, across the four subtypes, > 60% of the total sample number (therefore ≥ 209, since there are 348 samples in this dataset)); c) their associated q-value (according to Storey's method[54], for "calculable" and "relevant" p-values) < 0.01; (d) their median value is strictly greater than the first octile value of the ranked set of median values of the full gene list; (e) their standard deviation strictly belongs to the top 50% across all genes.

**Analysis of RNA-Seq data: genomic dissimilarity among subtypes**. After performing the gene selection above described, it was assessed that the 'genomic dissimilarity' for RNA-Seq data between the two subtypes located at the two sides of this heat map (i.e., Classical and Secretory) is the maximum among the six possible cases (one-to-one comparisons among four subtypes). This genomic dissimilarity was measured as follows: (1) for each gene $g_i$ and each subtype $s_j$ it was calculated the mean, named m($g_i$, $s_j$) = m$_{i,j}$. So, for $j$ that belongs to {Classical (C), Basal (B), Primitive (P), Secretory (S)}, the four vectors considered are: {$m_{i,C}$}, {$mi_B$}, {$m_{i,P}$}, and {$m_{i,S}$} (with $1 \le i \le 4291$, since the heat map of hybrid differential expression has 4291 genes); (2) the six possible city-block distances between the vectors of the four subtypes were calculated. For example, the city-block distance between {$m_{i,C}$} and {$mi_B$} is: d({$m_{i,C}$}, {$m_{i,B}$}) = $\Sigma_{1 \le i \le 4291}$ |$m_{i,C}$ - $m_{i,B}$|; (3) the maximum among these six city-block distances, which corresponds with the couple Classical-Secretory, was used to establish that these two subtypes have the largest genomic dissimilarity. This analysis confirmed that, also in this data subset: (a) the closest subtype to Classical, using this subtype-to-subtype distance, is Basal, and (b) the closest subtype to Secretory is Primitive.

**Analysis of RNA-Seq data: visualization of the clustered expression matrix**. Selected genes were log2 transformed, mean centered, hierarchically clustered (similarity metric: correlation (uncentered), clustering method: average linkage) using the version 3.0 of Cluster (open source clustering software)[55,56] and visualized with the Java-based program TreeView[57], in order to assess similarities and differences among their expression patterns. Here and in other heat maps utilized, genes with a null normalized expression value are log-transformed and, therefore, are assigned to a Not-a-Number (NaN) value, which TreeView displays

as a gray rectangle. The hierarchically clustered expression matrix, with respect to the genes (matrix rows) has 4,291 rows and 348 columns.

**Analysis of RNA-Seq data: definition of the upper and lower portion of the clustered expression matrix.** Keeping the four LUSC subtype in this order (i.e., (1) Classical, (2) Basal, (3) Primitive, (4) Secretory, from left to right) grants an interesting computational feature: namely, after splitting the matrix of gene expression into two halves (each with 2145 genes, so excluding the gene # 2146, which is exactly in the median position) and (i) calculating how many values are, for each gene and each subtype, above the mean for that gene, as a percentage and (ii) calculating the average of these means for the first half and the second half, separately, for each subtype, it turns out that in the upper half this value grows from left to right, while in the lower half it drops from left to right. This defines a clear gradient of gene expression and, for this reason, as well as for the genomic assessments, the above-mentioned subtype order was permanently selected. Due to the use of these data for multiple computational tasks, each independent of the other, the sample order inside each subtype was defined at the beginning of our analyses according to the sequence found in the original TCGA files (that relies on the TCGA sample barcodes) and to the availability of sample subtypes. Notably, this sample order was changed only in our analyses concerning the immunome signature (see section "Analysis of the immune cell types: scoring of different infiltrates"). In order to perform a batch analysis of survival data (see section "Survival data analysis based on tumor subtypes, clinical stage and expression levels for selected genes") for selected genes associated with hyper-expression (see below) in PS, we focused on the intermediate portion of the HM (precisely, from gene 1900 to 2400) and looked for a gene that could be algorithmically defined as the last of the upper HM portion. Specifically, for every gene we (1) assessed the percentage of samples hyper-expressing that gene (i.e., having an expression, for that gene, $\geq$ gene mean across the 348 samples) per subtype, (2) calculated the average of these two percentages for CB (Average(%C, %B)) and PS (Average(%P, %S)), and (3) determined the difference between these two values ($\Delta$(CB, PS) = Average(%C, %B) - Average(%P, %S)). Since when $1900 \leq$ (gene order # in HM) $\leq 2273$ it follows that $\Delta$(CB, PS) < 0 and when $2274 \leq$ (gene order # in HM) $\leq 2400$ it follows that $\Delta$(CB, PS) > 0, except for patterns of maximum three consecutive genes that do not meet the inequality requirement for $\Delta$(CB, PS) in the defined range of genes, we included in the batch analysis for survival all genes from 1 to 2274 (upper portion of RNA-Seq HM for survival analysis purposes). Therefore, both the maximum of the first interval (2273) and the minimum of the second interval (2274) were used for this analysis. Notably, the precise gene where the HM is split turned out to be not critical, since the last gene having statistical relevance for survival in the upper HM portion is gene #2261 (TKTL1).

**Analysis of RNA-Seq data: gene-based statistical analyses.** The statistical significance (1) of RNA-Seq data of the genes CD14 and F13A1 (presence of samples above and below the median for each subtype) was calculated using the binomial test (R); (2) for comparisons of gene expression levels across the four LUSC subtypes of the RNA-Seq data for the genes CD14, CCL2, CCL3, CSF1, and TNFα was assessed using one-way analysis of variance (ANOVA) on log-transformed data; (3) of the correlations (measured by the Pearson's coefficient) between CD14 and CCL2, CCL3 and CSF1 was calculated using a t-statistic t*, with 346 ( = 348 − 2) degrees of freedom. The formula used for t* is: $t^* = [r \cdot \mathrm{sqrt}(n-2)]/[\mathrm{sqrt}(1-r^2)]$, where $r$ is Pearson's correlation coefficient (calculated on non-transformed data), $n$ is the number of samples (348) and sqrt stands for "square root". For this computational step, we exploited the relatively large number of subjects of the TCGA LUSC dataset. These results were confirmed also using the Spearman's rank correlation coefficient. Finally, the p-values of point 1 (two sets of p-values) and of the genes CD14, CCL2, CCL3, CSF1 of point 2 (1 set of p-values) were used for calculating the corresponding false discovery rate (FDR) according to Benjamini and Hochberg[58]. The ANOVA test of point 2 was used assuming its robustness as for the normality requirement[59] and checking the data homoscedasticity across the four subtypes, for each gene. For this task, we used the Brown–Forsythe test (p-value threshold: 0.05) and the criterion of Dean and Voss about the ratio between the maximum and minimum variance among the subtypes[60]. Additionally, the results of the ANOVA test for these five genes were confirmed through the Kruskal–Wallis test.

**Gene ontology analysis of RNA-Seq data.** With the aim to be more selective for gene ontology (GO) purposes, we changed the point (e) of the section "Analysis of RNA-Seq data: identification of a set of representative genes" into the following point e': the standard deviation of selected genes has to strictly belong to the top 1/3, in the set of genes originally considered (i.e., 20,531 genes). The smaller list of genes so obtained (2972 genes total) was re-clustered as described (see section "Analysis of RNA-Seq data: visualization of the clustered expression matrix") and further split into a top (1423 genes) and bottom (1549 genes) portion, checking which was the 'splitting point' in the sequence of averages (calculated excluding missing values, which correspond to null values in the untransformed gene expression matrix) of log2 transformed, mean centered by gene and hierarchically clustered values in the Secretory subtype. We define as a "splitting point" in a sequence of real numbers a couple of consecutive values of that sequence such that they have

opposite sign (positive and negative or vice versa) and such that the sequence does not revert back to its previous sign for at least five couples of consecutive values following that "splitting point". Moving from top to bottom in the HM, when we detected a "splitting point", we included the last value with a positive sign (first gene of the "sliding couple") as the last gene of the Top portion (that mostly contains genes having positive averages) and the following genes were all attributed to the HM Bottom portion (that mostly contains genes having negative averages). Notably, this simple mathematical rule splits the HM into two identical parts either using the Secretory subtype or the PS aggregate as a reference (i.e., the "splitting point" of the sequence of averages does not change going from Secretory to PS). For this reason and for brevity, we refer to genes significantly up-regulated shown in the HM Top portion as genes characterizing the Secretory subtype, and to genes significantly up-regulated displayed in the HM Bottom portion as genes of the Classical subtype. The GO analysis that was selected relies on the Expression Analysis Systematic Explorer (EASE) score (a p-value obtained through an adjusted Fisher's exact test)[61] and was performed using DAVID Bioinformatics Resources[62]; the selected background was "Homo sapiens". Each GO category was considered for further analyses only when fulfilled these three criteria: (1) is referred to GO biological processes (BP); (2) has two or more gene members inside the list of genes of the Top HM portion; (3) has a p-value < 0.001.

**Ingenuity pathway analysis of RNA-Seq data.** Using ingenuity pathway analysis (IPA) (http://www.ingenuity.com/products/ipa) of the genes ($n = 403$) that are generally hyper-expressed (see the definition of hyper-expression given in section "Analysis of RNA-Seq data: definition of the upper and lower portion of the clustered expression matrix") in the Secretory subtype (more precisely, of the genes shown in the portion of the RNA-Seq HM defined as "upper" in section "Analysis of RNA-Seq data: definition of the upper and lower portion of the clustered expression matrix") and also have statistically significant p-values ($< 0.05$) for overall survival (see section "Batch survival data analysis based on gene levels"), we assessed for the most significant biological themes in terms of "Disease or Function". The displayed network is focused on the most significant biological function ("Leukocyte Migration") and shows all genes belonging to it. The levels of statistical significance for these genes (in terms of survival analysis) are displayed on a color scale (gray: least significant, red: most significant); genes are clustered (using colored ovals) according to specific leukocyte categories (e.g. IMs, Macrophages) based on previously described immune subset markers[27]. Using the same gene set ($n = 403$), we also utilized IPA to perform the "upstream regulator analysis" (URA) to identify which were the most significant upstream regulators inside this gene network. We utilized the Cytoscape software for the network visualization of IPA results[63]. Finally, the sub-cellular and extra-cellular 'locations' of the genes shown in the URA graph were assigned using IPA software.

**Survival data analysis based on tumor subtypes, clinical stage and expression levels for selected genes.** The survival of distinct experimental groups according to subtype, clinical stage and expression levels for selected genes was assessed using the log-rank test (a.k.a. Mantel-Cox test)[64]. These tests had 1 degree of freedom (df) for comparisons of two groups, and 3 for comparisons of four groups. When the comparison was based on expression levels, it was either performed using the median as a splitting point (see below) or lower, middle, and upper quartile (here intended as dividing points for the population). All the comparisons involving (a) subtypes, (b) subtype aggregates (see section "Analysis of RNA-Seq data: a spectrum of gene expression among the subtypes"), (c) clinical stages, (d) CD14, CCL2, CCL3, CSF1, and TNFα were analyzed using the computational implementation of this statistical test that is built-in in GraphPad Prism (http://www.graphpad.com/scientific-software/prism/). The FDR[58] was used for the multiple hypothesis testing correction of the statistical tests concerning CD14, CCL2, CCL3, and CSF1.

**Batch survival data analysis based on gene levels.** A broader screening of survival values for differentially expressed genes in the above-mentioned LUSC subtypes (see section "Analysis of RNA-Seq data: identification of a set of representative genes") was achieved through an in-house MATLAB (https://www.mathworks.com/products/matlab/) script that calculates the hazard ratios (HR) and log-rank test p-values using the "coxphfit" function for each gene selected[65] (see the script contained in the file survival_analysis.docx). This analysis, on a gene by gene basis, splits samples depending on their being above or below a specific threshold; then, HR are calculated for two sample populations, one defined as {Samples whose gene expression is $\geq$ gene median} and one defined as {Samples whose gene expression is < median}.

**Modified immunome signature.** Based on the paper of Bindea et al.[27], which describes "a compendium of mRNA transcripts" of genes whose expression is strongly associated with specific immune cell types, we put together a list of genes for which the following variables are known: (1) type of immunity (innate or adaptive), (2) immune cell type(s) (cell type(s) of the immune system characterized by that gene); (3) gene aliases. This gene list was made more comprehensive by adding selected genes of known function of the immune system and that were not already present in this file (IMs: CD14, CCL2, CCR2, CCL3, CSF1R, CSF1; M2

macrophages: TGFB, VEGFA, IL10, CD206, VCAM1, CD163, ICAM1, IL1RA, CSF3R; M1 macrophages: TLR2, TLR4, CD80, CD86, CCR7, CCL5, CXCL9, CXCL10, CXCL11). Finally, we completed this list by adding other immune-related genes described by Charoentong et al.[28], who build upon the work of Bindea et al.[27], for the following immune cell types: regulatory T-cells, activated DCs, myeloid-derived suppressor cells, neutrophils, and plasmacytoid DCs. The matching between gene identifiers of the immunome signature and of RNA-Seq data was based on Entrez gene identifiers (https://www.ncbi.nlm.nih.gov/sites/gquery); immune-related genes without a match based on these identifiers were discarded. Notably, at the end of this procedure, each gene of the modified immunome signature (598 total) belongs either to one or two immune cell types (572 genes have a unique and 26 have also a second immune cell type).

**Analysis of the immune cell types: scoring of different infiltrates.** Due to the importance of CD14 for the survival of LUSC patients (as an unfavorable prognostic factor), we generated analyses of the mRNA levels of 29 immune cell types where (i) samples were preliminarily ordered according to the expression levels of CD14 and (ii) immune cell type markers (i.e., the genes identified in the previous section) were used to assess the levels of immune cell type density. Specifically, in order to understand the role played by heterogeneous immune infiltrates, we created a scoring system, which determines the immune cell type density of each sample using the immune cell type markers, through the following steps: (1) the expression values of the immune cell type markers are, individually, ranked from the lowest to the highest levels using a score that goes from 1 to 348 (total number of samples analyzed); (2) when there is a tie (equal expression value of a gene in n samples) the final rank for the tied samples is the average among these n ranks, similarly to what happens in the Wilcoxon signed-rank test[66]; (3) for every sample, these ranks are averaged across the immune cell type markers (i.e., by columns of the expression matrix) of the chosen immune cell type, consistently with the previous point; (4) a cell type density score is assigned to each sample by using these rank averages. Of course, this procedure (a) assigns to each sample as many immune cell density scores as the analyzed immune cell types and (b) is independent of the preliminary CD14-based reordering of the samples, which is instead used for assessments concerning the survival of the immune cell types. Additionally, this ranking procedure is unambiguous, with the only exception of two samples, which have the same normalized RSEM expression levels of CD14. However, the presence of these two samples does not introduce any relevant bias in the analyses here described, since both patients belong to the group that over-expresses CD14 and even share the same subtype (Classical). Altogether, this semiparametric procedure (steps 1–4) is similar to what was previously described[67] for calculating immune gene signatures, but is preceded by the non-parametric steps 1 and 2. This approach is different from deconvolution methods such as CIBERSORT[68] and TIMER[69] because, in theory, with our algorithm, a sample can be relatively enriched with respect to the broadest range of immune cell types (from none to all). It also differs from ssGSEA[70], since it assesses the under- and over-representation of the genes that belong to a gene set working across the samples (i.e., relatively to them).

**Analysis of the immune cell types: computational visualization of the CD14 + infiltrates.** Individual genes of the 9 CD14 + populations (i.e., aDC, DC, iDC, IMs, M1, M2, Macrophages, MDSC and Neutrophils) were also hierarchically clustered, after the gene expression values were log2 transformed and median centered (see section "Analysis of RNA-Seq data: visualization of the clustered expression matrix"); genes belonging to two distinct immune cell types were used for the HMs of both.

**Analysis of the immune cell types: correlations among immune infiltrates and survival analysis based on the density of the CD14 + cell types.** Immune cell type density scores are used for two main purposes: (i) evaluating the level of correlation between the 29 immune cell types, in order to understand their coordinated biological action in these patients; (ii) calculating, for the 9 CD14 + populations, the differential in survival of samples having "high" (≥ median) vs. "low" (< median) cell type density scores. The correlations among cell types are measured using the Spearman correlation coefficient and displayed in a heat map where these coefficients, after being hierarchically clustered in both dimensions (i.e., across matrix rows and columns) without neither data transformation nor row/column centering, are coded by a gradient of colors whose extremes are, respectively, minimum and maximum of the entire matrix of correlation coefficients. Clearly, this heatmap/matrix $a_{i,j}$ is symmetrical with respect to its main diagonal ($a_{i^*,j^*} = a_{j^*,i^*}$). The above-mentioned survival analysis was based on the log-rank test and, for every test, p-value, HR and ±95% confidence interval (CI) of the HR were calculated. Thereafter, (i) p-values and (ii) CD14 scores vs. HRs were separately plotted. We have defined the CD14 scores of these nine cell types as the rank of the ratios between the average cell type density score in the high vs. low CD14 groups of samples (with respect to the median expression level of CD14 itself). These scores are used to assess the level of density of each cell type as a function of CD14 expression. Specifically, $y = 9$ when this ratio is the highest and $y = 1$ when this ratio is the lowest. Notably, HR > 1 for each of these nine cell types, since when their density is "high" the prognosis, collectively speaking, is always less

favorable (even though sometimes in a minor or negligible way, like in particular it happens for the M1 macrophages) than for LUSC patients, who have a "low" density of these immune cell types.

**Analysis of gene sets.** In order to deal with the level of 'sparsity' of the gene expression matrix and with the specific type of signal of TCGA RNA-Seq data, a number of gene expression rows were trimmed from the expression spreadsheet of LUSC patients before running a gene-set based analysis; this pre-processing step prevents the inflation of positive results. Specifically, all those genes either (i) that had an average expression value strictly in the lowest 1/8 of the dataset or (ii) that had a standard deviation strictly in the lowest 1/8 of the dataset, or iii) for which samples having a signal (defined as a value ≥ 0.5) were strictly in the lowest 1/8 of the dataset, were disregarded. Then, Gene Set Enrichment Analysis (GSEA)[71] was run using this trimmed matrix of gene expression (with 17,398 genes) after splitting (inside the categorical class (cls) file) the samples of the four subtypes into the two groups CB (208 samples total) and PS (140 samples total). As previously mentioned, these two aggregates were created based on the correlations among their genes and ordered based on the gene expression gradient of their constitutive subtypes (see section "Analysis of RNA-Seq data: definition of the upper and lower portion of the clustered expression matrix"). GSEA was directly run using the gene symbol identifiers and the number of sample label permutations was set at 1,000. Additionally, the cutoff thresholds for gene set sizes were 15 and 500, respectively at the upper and lower end, and the 'metric' used for ranking the genes was the signal-to-noise (S2N). The gene matrix transposed (gmt) file, which was curated by the Molecular Signatures Database (MSigDB) (http://software.broadinstitute.org/gsea/msigdb/) and allows defining the gene sets that are tested for enrichment, is C7-Immunologic Signatures v.5.0; this file contains 1,910 immune-related gene sets. Due to the exclusion values above defined and to the sizes of these gene sets, all of them were included in the analysis. The enrichment score (ES) of a gene set measures the level of enrichment found in the ranked list for that gene set; for our analysis, the ranking gave the highest priority to the genes of the PS aggregate. We considered statistically significant gene sets having a FDR < 0.05; this significance threshold is considerably lower than the value (i.e., 0.25) originally suggested by the Authors of the GSEA method and provides a high degree of selectivity. Among the selected gene sets, we performed a further refinement (context-based) according to their relationships with the biological findings described in this article. In consideration of the gene expression gradient of the four subtypes from Classical to Secretory and for the sake of brevity, the word Classical is (extensively) used, in the figures of this article, as representative of CB, while the word Secretory represents the PS ensemble.

**Microarrays-based analysis of the gene expression of a mouse model of lung squamous carcinoma.** The Affymetrix Mouse Gene 2.1 ST Array (http://www.thermofisher.com/us/en/home.html) was used for measuring the gene expression of a) normal murine bronchial epithelial cells (MBEC), b) the KLN205 murine lung squamous (parental) cell line, and c) its sub-clone LN4K1. The CEL file processing was performed using the Affymetrix Expression Console; background adjustment, quantile-normalization and summarization were accomplished using the Robust Multichip Analysis (RMA) algorithm[72]. Later, the set of RNA probes that are included in this array (41,345) was split into two distinct sets. The list of probes used for this analysis, together with the available Affymetrix annotation, is reported in Supplementary Data 12. After these bioinformatics steps, the differential analysis is based on two sequential computational procedures. In the first procedure, genes are selected when (i) are differentially expressed between MBEC (4 replicates) and KLN205 (three replicates), or (ii) are differentially expressed between MBEC and the metastatic-derived cell line LN4K1 (three replicates) or (iii) fulfill both (i) and (ii). At this stage, a gene is considered differentially expressed between two groups of replicates when a) its expression levels are strictly higher or lower than in the other group for each replicate used (combinatorics-based implied p-value: 0.0286); (b) the difference between the means of the two groups is ≥ 50%; c) its range is in the top 75% in the entire set of genes. The second procedure allows defining which genes follow a gradient of expression (growth or reduction) going from MBEC to KLN205 parental to the sub-clone LN4K1 (hence moving from "normal" to "primary tumor" to "metastasis"), so that this expression gradient is sustained across all the replicates of the three experimental conditions. For each gene, the growing pattern (pattern A) is sub-divided into two sub-patterns, namely A-1 and A-2, which are not mutually exclusive. The sub-pattern A-1 is based on the following requirements (which are extensively described for more easily allowing assessing the level of overlap with the sub-pattern A-2): (i) all the LN4K1 samples have a strictly greater expression than all the KLN205 samples; (ii) all the KLN205 samples have a strictly greater expression than all the MBEC samples; (iii) the average expression for the LN4K1 samples is strictly greater than the average expression for the KLN205 samples. The sub-pattern A-2 is based on the following requirements: (i) the expression of each LN4K1 sample is strictly greater than the average expression across all the samples considered ($10 = 4 + 3 + 3$); (ii) the expression of each KLN205 sample is strictly lower than the average expression across all the samples considered; (iii) the expression of each MBEC sample is strictly lower than the average expression across all the samples considered. Similarly, for each gene, the dropping pattern (pattern B) is sub-divided into two sub-patterns, namely B-1 and B-2, which are also not mutually exclusive. The sub-

pattern B-1 is based on the following requirements (extensively described, as for A-1): (i) all the LN4K1 samples have a strictly lower expression than all the KLN205 samples; (ii) all the KLN205 samples have a strictly lower expression than all the MBEC samples; (iii) the average expression for the LN4K1 samples is strictly lower than the average expression for the KLN205 samples. The sub-pattern B-2 is based on the following requirements: (i) the expression of each LN4K1 sample is strictly smaller than the average expression across all the samples considered; (ii) the expression of each KLN205 sample is strictly higher than the average expression across all the samples considered; (iii) the expression of each MBEC sample is strictly higher than the average expression across all the samples considered. Overall, a gene is selected when is differentially expressed and follows either gene pattern A or B. These selection steps generated a gene expression spreadsheet containing 2368 genes. Finally these genes are displayed in a HM, where selected genes are hierarchically clustered after their data have been log2 transformed and mean centered (see section "Analysis of RNA-Seq data: visualization of the clustered expression matrix").

**Determination of gene ontology biological processes shared between the LUSC mouse model and LUSC patients**. In order to understand which biological processes are potentially involved in LUSC progression while being micro-environment-independent, we used an analysis inspired by the work of Xu et al.[32]. Preliminarily, we discarded entries of the murine array without a gene symbol. Then, for every gene of the Affymetrix array that had multiple entries, we selected (a) one of them, indifferently, when the numerical values were identical for all samples, due to repeated annotation/listing of the same gene or (b) the entry having the highest standard deviation across these two sets of murine samples. Then, orthologous genes between mouse and human and that were defined as differentially expressed between MBEC and LN4K1 samples (see the previous section) were included in the matched list of TCGA LUSC differentially expressed genes when: (a) had a percentage of presence across the samples of the Classical and Secretory subtypes greater than 60%; (b) the Benjamini–Hochberg FDR[58] calculated on the Wilcoxon rank sum test p-values between Classical and Secretory (after passing the previous gene filter) was < 0.00005. The choice of these two subtypes was based on their highest level of genomic dissimilarity (see section "Analysis of RNA-Seq data: genomic dissimilarity among subtypes"). Genes of these two subtypes were split according to having the ratio median(Classical)/median(Secretory) above or below 1 and clustered, as for the TCGA LUSC RNA-Seq data, in two separate heat maps, as described in section "Analysis of RNA-Seq data: visualization of the clustered expression matrix". Then, the corresponding murine genes of the MBEC and LN4K1 samples of the Affymetrix arrays were aligned to the human genes and displayed through two additional heat maps (for ratios of the medians above and below 1 as well), using the visualization style adopted in the main analysis of our mouse model of lung squamous carcinoma (see the previous section). Genes of these two groups were used for two separate GO analyses (see section "Gene ontology analysis of RNA-Seq data"), with "Homo sapiens" as background species and selecting GO-BP terms whose p-values were < 0.001 and, at the same time, containing at least 10 genes of either of these two groups.

**ELISA assays**. Murine CCL2 protein levels were quantified by ELISA using the DuoSet Immunoassay kit (R&D Systems DY479-05 and DY008) according to the manufacturer's protocol. To assess secretion of CCL2 in vitro, 344SQ, KLN205, and LN4K1 were seeded at a density of 400,000 cells per well in 3 ml of media in 6-well plates. Supernatant was collected 48 h later and stored at −80 °C. For analysis of plasma CCL2 levels, blood (approximately 200 μL per mouse) from 3 to 5 mice per group was obtained 1-week prior to sacrifice via submandibular bleed using a Goldenrod lancet (4 mm). Blood was collected into Vacutainer Blood Collection Tubes with anti-coagulant. Tubes were centrifuged at 25,000g at 4 °C for 5 min, then plasma was collected and stored at −80 °C until assay. Samples were assayed in triplicate and data represents the mean concentration.

**Proliferation assays**. KLN205 and LN4K1 cells were seeded at a density of 25,000 cells per well in 6-well plates in triplicate and counted on a hemocytometer using a Trypan Blue counterstain.

**Monocyte fibrin cross-linking protocol**. Low (25k) and high (100k) density of monocytes were incubated with unfractionated or Peak 1 fibrinogen (Enzyme Research Laboratories, South Bend, IN), in the absence and presence of T101 (Zedira, Darmstadt, Germany) for 15 min at 37 °C. Clotting was triggered with thrombin (Enzyme Research Laboratories, South Bend, IN) and CaCl2 (1 U/mL and 10 mM, final, respectively) and clot formation proceeded for 2 h. Samples were then dissolved in 50 mM dithiothreitol, 12.5 mM EDTA, and 8 M urea at 60 °C for 1 h, diluted 120-fold in 6 × reducing SDS sample buffer (Boston Bioproducts, Ashland, MA), boiled, separated on 10% Tris-Glycine gels (Bio-Rad, Hercules, CA), and transferred to polyvinylidene difluoride membranes (Invitrogen, Carlsbad, CA). Membranes were blocked for 1 h at room temperature with Odyssey Blocking Buffer (LI-COR Biosciences, Lincoln, NE), incubated overnight at 4 °C with primary anti-human fibrinogen polyclonal antibodies (Clone A0080, Dako, Glostrup, Denmark), and then incubated with Alexa Fluor®-488 fluorescence-labeled anti-rabbit secondary antibodies (Immunoresearch, West Grove, PA) for 1 h at room temperature. Membranes were scanned on a GE Typhoon FLA-9000 Imager (GE Healthcare, Pittsburgh, PA). Fibrin(ogen) bands were quantified by densitometry (ImageJ 1.48 v). Band intensities of fibrin γ-γ dimers and high molecular weight cross-linked fibrin species were normalized to the fibrin(ogen) Bβ + β-chain before normalizing to time zero.

**Invadopodia assays**. LN4K1 cell lines (50,000) were mixed with 200 μg/mL Fibronectin, 2 mM CaCl2 with 2 mg/mL unfractionated fibrinogen or Peak 1 fibrinogen. Coagulation was induced with 2.5 U/mL thrombin by incubating at 37 °C for 30 min. Clot-embedded cells were incubated with MEM medium supplemented with 10% FBS at 37 C in cell culture incubator. The images were captured on clot-embedded cells at 24 h by using phase contrast or fluorescence microscopy. Elongated or stellate structures of cells were classified as invadopodia, whereas others (round shaped cells) designated as negative for the formation of invadopodia. Percent Invadopodia were calculated as invadopodia-forming cells divided by a total number of cells in each of six fields.

**Invasion assays**. LN4K1-GFP (50,000 cells) and bone marrow-derived IMs (100,000 cells) were mixed with 200 μg/mL Fibronectin, 2 mM CaCl2, and either 2 mg/mL unfractionated fibrinogen, Peak 1 fibrinogen or BD Matrigel (total final volume of 50 μL) on top of a Boyden 8 μm migration chamber. Coagulation was induced with 2.5 U/mL thrombin by incubating at 37 °C for 15 min. Clot-embedded cells were incubated with serum free MEM medium (100 μL), and MEM + 10% FBS was used as a chemoattractant in the bottom chamber. Invasion was assessed 24 h later. For experiments using T101, a concentration of 50 μM was used.

**Analysis of NFκB-dependent CCL2 induction**. LN4K1 cells were subjected to NF-κB subunit p65 silencing or IKKβ inhibition by transfecting with 25 nM p65-specific or non-specific control siRNA using Lipofectamine RNAiMAX (Invitrogen, Carlsbad, CA) or by treating with 5 μM Compound A, an IKKβ inhibitor, for 5 h. Twenty-four hours after siRNA transfection, the medium was replaced, after which the cells were incubated for an additional 24 h and then treated with 100 ng/ml recombinant TNF for 2 h. After all treatments, the cells were harvested in RNA lysis buffer and subjected to RT-qPCR analysis.

**Western blotting**. After treatments, cells were lysed by scraping in RIPA buffer (ThermoFisher, cat no. PI89901) containing 1 mM PMSF, 1 mM NaVO4, 1 mM dithiothreitol, and 1 × protease inhibitor cocktail. Equal amounts of lysates (20-30 μg of total protein) were run on 10% SDS-PAGE gels, after which protein was transferred to nitrocellulose membranes (BioRad, Hercules, CA). Membranes were blocked in 5% BSA/Tris-buffered saline-Tween 20 (TBS-T) for one hour at room temperature prior to probing with primary antibodies overnight at 4 °C. Primary antibodies included anti-phospho-p65 (Ser 536, clone 93H1, #3033) and anti-p65 (clone D14E12, #8242) from Cell Signaling Technology (Danvers, MA), and anti-vinculin (clone hVIN-1, #V9131) from Sigma. After probing with primary antibodies, membranes were washed three times in TBS-T and then probed with the appropriate horseradish peroxidase-conjugated secondary antibodies (anti-mouse (#115-035-003) or anti-rabbit (#111-035-003) from Jackson ImmunoResearch). Then, the membranes were washed four times in TBS-T and developed using Clarity Western ECL substrate (BioRad, #1705060). Membranes were visualized using a BioRad ChemiDoc MP system (BioRad, Hercules, CA).

**Flow cytometry**. Blood, Bone marrow and tumors were collected for flow cytometry analysis. Lung tissues were washed and mechanically minced using a sterile scalpel in low glucose DMEM and digestion media (1 mL collagenase at 2 mg/ml, and 15 μL DNase at 1 mg/mL). Tissue was digested into a single cell suspension by light shaking in digestion media for 30 min at 37 °C and was then filtered through a 40-μM cell strainer, pelleted, treated with ACK lysis buffer at room temperature for 2 min, and then pelleted again. Cells were re-suspend in FACS buffer (0.5% BSA and 2 mM EDTA in PBS) at a concentration of ~$10^6$ cells/100 μL. Samples were incubated with Fc block (10μL/100μL) on ice for 15 min and then with the following antibodies: CD45 (APC-conjugated, #103112 Biolegend), CD11b (PE/Cy5 conjugated, #101210 Biolegend), Ly6C (PE/Cy7-conjugated, #128018 Biolegend), NK1.1 (PE/Cy7-conjugated, #108714 Biolegend), Ly6G (PE-conjugated, #127608 Biolegend), CD25 (PE/Cy7-conjugated, #102016 Biolegend), CD49b (PE-conjugated, #108908 Biolegend), CD8a (FITC-conjugated, #100706 Biolegend), TCR β chain (PE-conjugated Cat#109208 Biolegend), Siglec-F (BV421-conjugated, #562681 BD bioscience), F4/80 (FITC-conjugated, #123108 Biolegend), MHCII (PE-conjugated, #107608 Biolegend), CD11c (PE/Cy7-conjugated, #117318 Biolegend), CD4 (APC/Cy7-conjugated Cat#100414, Biolegend), CD206 (BV785-conjugated, Biolegend, clone C068C2), gdTCR (BV605-conjugated, Biolegend, clone GL3) and LIVE/DEAD® Fixable Violet Dead Cell Stain (# L34963 ThermoFisher Scientific). Approximately 0.2 μg of antibody was used for every $10^6$ cells. Cells were incubated with antibody for 30 min on ice, in the dark. Cells were then washed two times with FACS buffer and taken for flow cytometry analysis on an LSRFortessa. The collected data were analyzed using FlowJo software V10. When performing FACS on lung tissues, Siglec-F positive cells were included with dead

cells to remove alveolar macrophages and/or eosinophils. IMs were identified as CD45 + /CD11b + /Ly6C$^{High}$/Ly6G- cells, RMs as CD45 + /CD11b + /Ly6C$^{Low}$/ Ly6G-, granulocytes as CD45 + /CD11b + /Ly6C-/Ly6G + and gated on CD11b. TMAs as CD45 + /Gr1-/TCRb-/SiglecF-/CD11b + /F480 + /MHCII + and CD206 was used to distinguish high and low subsets, DCs as CD45 + /Gr1-/TCRb-/ SiglecF-/CD11c + /F480-/MHCII + , natural killer cells as CD45 + /TCRb-/NK1.1 + /CD49 + , CD4 + T-cells as CD45 + /TCRb + /CD4 + /CD25-, CD8 + T-cells as CD45 + /TCRb + /CD8 + and regulatory T cells as CD45 + /TCRb + /CD4 + /CD25 + .

**Quantitative real-time PCR**. For mRNA quantification, total RNA was extracted from cells using the Quick RNA MiniPrep Zymo Research Kit (Genesee Scientific). Using 1000 ng of RNA, cDNA was synthesized using an iScript cDNA Synthesis Kit (Bio-Rad) as per the manufacturer's instructions. Analysis of mRNA levels was performed on a StepOnePlus Real-Time PCR System (Applied Biosystems). Specific primers for [CCL2 (murine) (forward)-AGCACCAGCCAACTCTCACT, (reverse)-TCATTGGGATCATCTTGCTG; CCL3 (murine) (forward)- CCTCTGTCACCTGCTCAACA, (reverse)-GATGAATTGGCGTGGAATCT; CSF-1 (murine) (forward)-CGAGTCAACAGAGCAACCAA, (reverse)- TGTCAGTCTCTGCCTGGATG; RelA/p65 (murine) (forward)- GCTCCTGTTCGAGTCTCCAT, (reverse)- TTTGCGCTTCTCTTCAATCC; F13a1 (murine) (forward)- GAGCAGTCCCGCCCAATAAC, (reverse)- CCCTCTGCGGACAATCAACTTA; VEGFa (murine) (forward)- AACGAT-GAAGCCCTGGAGTG, (reverse)- GACAAACAAATGCTTTCTCCG] were used for SYBR Green-based real-time PCR, and 18 s rRNA was used as a housekeeping gene. PCR was done with reverse-transcribed RNA, 1 μL each of 20 μM forward and reverse primers, and 2 ×PowerUp SYBR Green Master Mix in a total volume of 25 μL. TaqMan Assays (Applied Biosystems) were used for TNFa expression (Mm00443258), and GAPDH (Mm99999915_g1) was used as a housekeeping gene. PCR was done with reverse-transcribed RNA, 20 ×TaqMan probe, and TaqMan Universal Master Mix II as per the manufacturer's instructions. For both SYBR and TaqMan-based PCR, each cycle consisted of 15 s of denaturation at 95 ° C and 1 min of annealing and extension at 60 °C (40 cycles). Reactions were run in triplicate.

**mRNA microarray**. Total RNA was extracted from MBECs, KLN205, and LN4K1 cells using the Quick RNA MiniPrep Zymo Research Kit (Genesee Scientific). RNA purity was assessed by a Nanodrop (Thermo Scientific) spectrophotometric measurement of the OD260/280 ratio, with acceptable values falling between 1.9 and 2.1. The RNA integrity number (RIN score) was determined using an Agilent TapeStation 2200 with acceptable values considered to be above 7.5. Total RNA (250 ng) was used to synthesize fragmented and labeled sense-strand cDNA and hybridize onto Affymetrx arrays (Affymetrix Mouse Gene 2.1 ST 16-Array Plate (902139)). The Affymetrix HT WT User Manual was followed to prepare the samples. Briefly, the WT Expression HT Kit for Robotics (Ambion) was used to generate sense-strand cDNA from total RNA. Following the synthesis of sense-strand cDNA, the cDNA was fragmented and labeled with the Affymetrix GeneChip HT Terminal Labeling Kit. The Beckman Coulter Biomek FXP Laboratory Automation Workstation with the Target Express set up was used to prepare the samples with these two kits. Fragmented and labeled cDNA was used to prepare a hybridization cocktail with the Affymetrix GeneTitan Hybridization Wash and Stain Kit for WT Arrays. Hybridization, washing, staining and scanning of the Affymetrix peg plate arrays was carried out using the Affymetrix GeneTitan MC Instrument. Affymetrix GeneChip Command Console (AGCC) Software was used for GeneTitan Instrument control.

**Immunostaining**. Staining was performed in formalin-fixed, paraffin embedded tumor sections (8 μm thickness). After deparaffinization, rehydration and antigen retrieval, 3% $H_2O_2$ was used to block the endogenous peroxidase activity for 10 min. Protein blocking of non-specific epitopes was done using 5% normal horse serum + 1% normal goat serum in TBS-T for 20 min, 2.8% fish gelatin in TBS-T minutes, or for monoclonal mouse anti-mouse antibodies, an Avidin/biotin kit (Vector Lab SP-2001) and Vector MOM immunodetection kit (BMK-2202) were used for blocking. Slides were incubated with primary antibody for KRT5 (rabbit anti-mouse, 1:500, Dako Z0622), p63 (monoclonal mouse anti-mouse, 1:100, Biocare CM 163), CD-31 (rat anti-mouse, 1:400, Pharmingen 553370), TTF-1 (monoclonal mouse anti-mouse, 1:200, Dako M3575), or Ki-67 (rabbit anti-mouse, 1:200, Abcam ab15580) overnight at 4 °C in blocking solution. After washing with PBS, the appropriate amount of horseradish peroxidase-conjugated secondary antibody was added and visualized with 3,3′-diaminobenzidine chromogen and counterstained with Gill's hematoloxylin #3. Light field images were obtained using a Nikon phase microscope. To quantify microvessel density (MVD), we examined 5–10 random fields at 100 × magnification for each tumor (5 tumors per group) and counted the microvessels within those fields. A vessel was defined as an open lumen with at least one adjacent CD31-positive cell. Multiple positive cells beside a single lumen were counted as one vessel, and quantification was performed by two investigators in a blinded fashion. Proliferation indices were determined using three representative fields at 200 × magnification for each tumor (5 tumors per group). All Ki-67 positive cells per high-powered field were enumerated. Ki-67

expression was analyzed using CellProfiler 2.0 software[73] to quantify the number of positively staining cancer cells per high-powered field (200 × magnification). Tissue microarray samples for lung squamous cell carcinoma cancers (previously categorized according to mRNA subtype)[8] were obtained and prepared following institutional review board approval for UNC Chapel Hill. Working with UNC's tissue pathology laboratory, the tissue microarrays were stained for CD14 (Leica Biosystems: mouse monoclonal anti-human CD14, #NCL-L-CD14-223, clone 7) and cross-linked fibrin (Zedira, mouse anti-human cross-linked fibrin, A076, 1:1,500 dilution). For multiplexed staining, rabbit monoclonal antibody against CD14, clone EPR3653, # 114R-14 was from Cell Marque (Rocklin, California), mouse monoclonal antibodies were: anti-CCR2, clone 7A7, # ab176390, (Abcam, Cambridge, MA), anti-CK, clone AE1/AE3, # M3515 (Agilent Technologies/ DAKO, Santa Clara, CA), anti-CD206, # ab64693, (Abcam, Cambridge, MA) and anti-D-Dimers (cross-linked fibrin), # A079 (Zedira GmbH, Darmstadt, Germany). Single IHC and triple IF (3plex IF) stains were carried in the Leica Bond-III fully automated staining platform (Leica Biosystems Inc., Norwell, MA). Slides were dewaxed in Bond™ Dewax solution (AR9222) and hydrated in Bond Wash solution (AR9590). Epitope retrieval for all targets were done for 20 min in Bond-epitope retrieval solution 1 pH6.0 (AR9661). The epitope retrieval was followed with 5 min endogenous peroxidase blocking using Bond peroxide blocking solution (DS9800) and 10 min protein blocking only for CK. For the 3-plex CD14-CCR2-CK immunofluorescence stain the application order and incubation times of the primary and secondary antibodies and the TSA systems were the following: (1) CD14, 1:200, 2 h, Bond polymer (DS9800) 8 min, Tyramide Reagent Alexa Fluor™ 488 (1:50) 15 min (#B40953, Life Technolonogies), (2) CCR2- 1:400, 1 h, Bond post primary (DS9800)- 8 min, Bond polymer-8 min, and TSA-Cy5 (1:50)-15 min (#SAT705A001EA, Perkin Elmer), and (3) CK- 1:500, 1 h, Bond post primary- 8 min, Bond polymer- 8 min, and TSA-Cy3 (1:50)-15 min (#SAT704A001EA, Perkin Elmer). Between the stains the appropriate antigen retrieval (10 min) and peroxide blocking steps were inserted. Stained slides were counterstained with Hoechst 33258 (#H3569, Life Technologies) and mounted with ProLong® Diamond Anti-fade Mountant (P36961, Life Technologies). Single stain controls were done for 3plex IF when one primary antibody was omitted to make sure that cross reactivity between the antibodies did not occur. For single IHC stain D-Dimers antibody (1:1500) was applied for 30 min and detection was done using Bond™ Polymer Refine kit with 3,3′-diaminobenzidine (DAB) visualization and Hematoxylin counterstain (DS9800). Stained slides were dehydrated and coverslipped. Positive and negative controls (no primary antibody) were included for IHC and IF stains. IHC were digitally imaged in the Aperio ScanScope XT (Leica Biosystems Inc., Norwell, MA) using 20 × objective. High resolution acquisition of CD14-CCR2-CK IF slides in the DAPI, AF 488, Cy3 and Cy5 channels was performed in the Aperio ScanScope FL (Leica) using 20 × objective. Nuclei were visualized in DAPI channel (blue), CD14 in AF 488 (green), CK in Cy3 (cyan) and CCR2 in Cy5 (red). Using Aperio software, following color deconvolution, a previously described IHC scoring criteria using Aperio software was utilized to obtain H-scores for CD14 and cross-linked fibrin expression[74]. For automated scoring of multiplexed images, slides containing fluorescently labeled TMAs sections were scanned in the Aperio ScanScope FL (Leica Biosystems) using 20 × objective and images were archived in TPL's eSlide Manger database (Leica Biosystems). Cytokeratin staining was used to digitally separate tissue cores into cytokeratin positive and negative regions (Tissue Studio Composer; Tissue Studio version 2.5 with Tissue Studio Library version 4.2; Definiens Inc., Carlsbad CA). Automated digital analysis of individual tissue cores was run separately in these two regions. Tissue Studio software, specifically the Nuclei and Simulated Cells algorithm in the IF Portal, was then used to detect and enumerate cells that co-expressed biomarkers of interest in the annotated regions. Briefly, nuclei were digitally detected by the presence of Hoechst stain (nuclear counterstain). From these nuclei, a cell simulation was performed – cells margins were grown out from nuclear boundaries. For this dataset, positivity thresholds for CD14 + and CCR2 + were determined by measuring the average staining intensities both inside and outside simulated cells. Once thresholds were set, the algorithm evaluated each cell individually for the presence of CD14 and CCR2. Cells that were negative for both markers or positive for CD14, CCR2 and both CD14 and CCR2 were enumerated by the algorithm.

**Immunocytochemistry**. Cells were centrifuged at 1,000 rpm for 5 min in a Cytospin 3 (Shandon). Cells were then fixed with 4% PFA for 15 min and permeabilized with 0.25% Tween 20 in PBS for 15 min at RT. Protein blocking was done with 2% BSA and 0.25% Tween for 1 h at RT. Slides were incubated with primary antibody for CD11b (rabbit, 1:100, Abcam ab133357) and/or FXIII (sheep, 1:100, Enzyme Research Labs SAF13A-AP) in blocking buffer at 4 °C for 16 h. After washing cells were incubated with appropriate secondary antibodies, goat anti-rabbit (Alexa Fluor 488) and/or goat anti-sheep (Alexa Fluor 594), diluted 1:500 in blocking buffer for 1 h at RT. Hoechst (1:10,000) was used for nuclear staining. Coverslips were mounted with Prolong Gold (Invitrogen). A Leica DMi8 inverted microscope was used for fluorescent micrography. A Zeiss 710 confocal microscope was used for confocal imaging. All image processing was done with FIJI software.

**Statistical analysis for experiments and tissue microarrays**. Between 5 and 10 mice were assigned per treatment group; this sample size gave approximately 80% power to detect a 50% reduction in tumor weight with 95% confidence. Results for

each group were compared using Student t test (for comparisons of two groups) and analysis of variance (for multiple group comparisons). For values that were not normally distributed (as determined by the Kolmogorov-Smirnov test), the Mann–Whitney rank sum test was used. A P-value less than 0.05 was deemed statistically significant. Aggregated data of Supplementary Figure 4 were analyzed through a two-tailed binomial test (CD206 + vs. CD206- cells in the population of CD14 + /CCR2 + cells), with an expected frequency of 0.5 for each 'trial', using R. All other statistical tests for in vitro and in vivo experiments were performed using GraphPad Prism 7 (GraphPad Software, Inc., San Diego, CA). The multiple hypothesis testing correction of these results was made using the FDR[58].

**Online content**. Supplementary and Source Data are available in the online version; references unique to these sections appear only in the online version.

**Data availability**. The Affymetrix microarray data that support the findings of this study have been deposited in the Gene Expression Omnibus (GEO) data bank, accession code GSE112585.

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

## Acknowledgements

The authors would like to especially thank Drs. Stephanie Cohen, Nana Feinberg and Mervi Eeva from the UNC Translational Pathology Lab for their help with immunohistochemistry and processing of the tissue microarrays, the UNC Animal Histopathology Core, and Janet Dow from the UNC Flow Cytometry Core Facility. The authors thank Dr. Antonio Amelio for providing the Fluorescent-Nanoluciferase plasmids. The UNC Translational Pathology Laboratory is supported in part by grants from the NCI (2-P30-CA016086-40), NIEHS (2-P30ES010126-15A1), UCRF, and NCBT (2015-IDG-1007). C.V.P. was supported in part by the National Institutes of Health R01CA215075, a Mentored Research Scholar Grants in Applied and Clinical Research (MRSG-14-222-01-RMC) from the American Cancer Society, the Jimmy V Foundation Scholar award, the UCRF Innovator Award, the Stuart Scott V Foundation/Lung Cancer Initiative Award for Clinical Research, the University Cancer Research Fund, the Lung Cancer Research Foundation, the Free to Breathe Metastasis Research Award and the Susan G. Komen Career Catalyst Award. S.H.A. was supported in part by a grant from the National Institute of General Medical Sciences under award 5T32 GM007092. E.B.H. was supported in part by a grant from the National Cancer Institute of the National Institutes of Health under award number T32CA196589. S.H.R. was supported in part by NIH P30DK065988. L.M. was supported in part by the University Cancer Research Fund and NCI CA180134. A.B. was supported by NIH R35CA197684. A.S.W. was supported in part by NIH R01HL126974. S.K. was supported in part by NIH F31HL139100. The UNC Flow Cytometry Core Facility and Lineberger Comprehensive Cancer Center Animal Histopathology and Animal Studies Cores are all supported in part by an NCI Center Core Support Grant (CA016086) to the UNC Lineberger Comprehensive Cancer Center. The UNC Flow Cytometry Core Facility is also supported in part by the North Carolina Biotech Center Institutional Support Grant 2012-IDG-1006.The views expressed in this article are those of the authors and do not reflect the official policy of the Department of Defense or U.S. Government.

## Author contributions

Conception and design: C.V.P.; Development of methodology: A. Porrello, E.H.H, P.L., A.H., A.B., D.N.H., C.O., S.K., A.S.W., C.V.P.; Acquisition of data (provided animals, acquired and managed patients, provided facilities, etc.): C.V.P., S.K., A.S.W., A.P., J.H., S.H.A., E.H.H., P.L., S.B., S.K.G., T.A.W., S.H.R.; Analysis and interpretation of data (e.g., statistical analysis, biostatistics, computational analysis): A.P., E.H.H., P.L., S.H.A., S.K.G., A.H., A.E., M.D.W., T.A.W., S.K., A.S.W., C.V.P.; Writing, review, and/or revision of the manuscript: All authors, Administrative, technical, or material support (i.e., reporting or organizing data, constructing databases): C.V.P., A.P., M.W.; Study supervision: C.V.P.

## Additional information

**Competing interests:** C. Oderup is an employee of Pfizer, Inc. The other authors disclosed no potential conflicts of interest.

