## [Peer Review File · Nature Communications]

Reviewers' comments:

Reviewer #1 (Remarks to the Author):

The authors convincingly use patient data from the TCGA to highlight the elevated presence of CD14+ myeloid populations as a factor for poor survival in LUSC. The authors propose a mechanism whereby this cell population contributes to metastatic progression through Factor XIIIa. The study is interesting and presents new data in the field. Issues to address in order to elevate the rigor and impact of the study are listed below.

(1) The findings are somewhat surprising since it is more commonly thought that recruited monocytes differentiate into tumor-associated macrophages and MDSC, and these cells are more associated with pro-tumor activities. One possibility could be that only a subtype of CD14+ CCR2+ myeloid cells, perhaps in a mid-differentiation state from inflammatory monocytes and mature macrophages/DCs, are associated with survival. Further characterization of the CD14+ CCR2+ cells with additional markers could be necessary as they may not fit into the classical monocyte/M1/M2 classification.

(2) The authors hypothesized that these myeloid cells could be enhancing the metastatic properties of the cancer cells. Although they compare secretion of TNF-alpha and CCL2 between the KLN205 and LN4K1 lines, they do not show that parental LN4K1 induce higher IM recruitment than parental KLN205 (only comparing with non-cancer cells). This data needs to be shown to validate the hypothesis that higher IM count leads to increased metastatic capabilities between the 2 cell lines. As the authors mentioned in the paper, other chemokines could also be responsible for recruitment. This concern is also supported by the fact that injection of CCL2-overexpressing KLN205 doesn't lead to a much higher IM count in the lungs compared to healthy mice (comparing Fig. 5e to Fig. 4j) or mice receiving CCL2-knockdown LN4K1 (Fig. 5e vs 5g), and yet leads to significant survival differences. Further, the recruitment of inflammatory immune cells to the lungs simply due to orthotopic injections should be addressed.

(3) The in vivo experiments fail to show how IMs promote metastasis. Higher IM count could be independent from enhanced metastasis and could simply promote tumor initiation and growth, which will lead to enhanced metastasis. The following set of experiments also do not show a clear correlation between IMs and enhanced metastasis. Although the formation of invadopodia is shown across different conditions, migration and invasion assays are required to show the enhanced invasive capabilities of cancer cells under the various conditions. Since the authors show that IMs have a stronger correlation with poor survival than M2 macrophages (which is the polarization associated with TAMs), does FXIIIa expression decrease once differentiated to M2 similarly to resident macrophages? Authors should show that siRNA FXIIIa knockdown in IMs leads to diminished cancer cell invadopodia formation on Peak 1 fibrinogen; and that the T101 compound has no effect on invadopodia formation and invasion using a substrate other than fibrinogen.

(4) The authors do not show that blocking the Factor XIIIa-mediated mechanism that is proposed prevents metastasis in the orthotopic system in vivo.

(5) Many of the phenotypic effects are of limited effect size. This could be addressed by incorporating more models to show the robustness of the differences in phenotypes in the different experiments.

Reviewer #2 (Remarks to the Author):

There is currently a clinical need for targeted therapies in lung squamous cell carcinoma (LUSC). In this manuscript, Porrello et al. have set out to address this need by identifying a subset of LUSCs that are enriched for inflammatory monocytes (IMs) and demonstrate poor clinical

prognosis. Applying a previously reported molecular subtyping scheme to LUSC samples from TCGA, Porrello et al. first showed that compared to other subtypes (classical, basal, and primitive), the secretory subtype is characterized by overexpression of IM-related genes such as CD14 and CCR2, and poor survival. Further utilizing transcriptome data, the authors demonstrated that among various immune cells that express CD14, IM gene signature shows stronger correlation with CD14, and is more significantly associated with poor survival. Porrello et al. next established a syngeneic metastasis mouse model and showed that TNF α activation of NFKB promotes lung squamous CCL2 secretion and IM recruitment. Moreover, using gene overexpression and RNA silencing experiments, they concluded that CCL2 is necessary and sufficient to enhance LUSC tumor metastasis in this model. Finally, the authors proposed that IMs promote tumor progression and enhance metastasis in part via expression of Factor XIIIa, an enzyme of the blood coagulation system that crosslinks fibrin.

The manuscript by Porrello et al. aims to answer important questions regarding the heterogeneity of LUSC with respect to infiltration by immune cells, in particular myeloid cells, and highlights potential cell-intrinsic mechanisms associated with the observed heterogeneities. Overall, the manuscript is easy to follow and the findings are potentially relevant for development of targeted therapies in LUSC. However, several issues require clarification and/or experimentation, as outlined by figure # below.

1. Figure 1

1.a. The authors have used the nearest shrunken centroid (NSC) classifier employed by [Wilkerson et al., 2010 Clin Cancer Res] to assign subtypes to TCGA tumors. The original classifier developed by Wilkerson et al. was based on microarray data, whereas TCGA data is based on RNA-seq. Like most machine learning algorithms, the NSC classifier operates under the assumption that the training and test data will be drawn from the same distribution. Challenges associated with cross-platform comparison of gene expression data (RNAseq vs. microarray [Thompson et al., 2016 PeerJ]) or even between different microarray platforms [Shabaln et al., Bioinformatics 2008]) have been pointed out in the past. This raises doubt about the accuracy of subtype labels that Porrello et al. assigned to TCGA samples. The authors should provide evidence that the training and test data have similar distributions, or better yet apply the unsupervised consensus clustering approach of Wilkerson et al. to TCGA data to de novo determine transcriptional subtypes and compare their findings to that of Wilkerson et al.

1.b. In panels a, b, and d, how did the authors address multiple testing?

1.c. In panel c, it is not clear to me how the 4291 genes shown in the heatmap were identified. From the supplementary methods, it appears that the authors first performed some filtering based on variability, and then performed differential expression analysis. However, it is not clear how the authors addressed the mean-variance trend inherent to RNAseq data during variance filtering and differential expression analysis.

1.d. What does "hyper-expressed genes" mean? Is it synonymous to "significantly up-regulated genes"? If so, please stick to the latter to avoid confusion.

1.e. Regarding panel h, in the main text it is stated that several markers of IMs were leading predictors of progression (i.e. had large hazard ratios), however only 2 markers are listed in the figure. A more objective approach would be to compare the location of IM- and non-IM-associated genes on a ranked list of survival genes.

2. Extended Figure 1

2.a. In the main text, it is stated that LUSC subtypes were balanced for clinical features, whereas Wilkerson et al. have reported that tumor differentiation and patient gender were associated with

subtype. Did the authors check patient gender as part of clinical features in TCGA data? What's the possible explanation for this potential discrepancy?

3. Figure 2

3.a. In panels a, b, and c, how did the authors address multiple testing?

3.b. In panels d, e, and f, what is the unit of y axes (gene expression)? RSEM expected or normalized counts? Did the authors performed statistics on counts or log-transformed counts?

3.c. In panels g, h, and i, how are the authors confident that CD14+/CCR2+ cells are IMs and not TAMs?

4. Extended Figure 2

4.a. Adjustment for multiple comparisons seems necessary.

5. Figure 3

5.a. The analysis performed is strongly dependent on gene signatures. The authors should provide complete list of gene signatures used in their analysis as supplementary material.

5.b. Several studies (including those cited by Porrello et al.) have provided objectively selected gene signatures. In the supplementary methods, the authors have stated that the gene lists were made comprehensive by adding selected markers. The addition of CD14 (which is not a monocyte-specific marker), CSF1 and CSF1R (which are canonical TAM and DC markers), as well as cytokines of interest (CCL2/CCL3) to the IM signature seems to introduce bias in the results given that as shown in Figure 2 these genes are expected to be overexpressed in the secretory subtype.

5.c. How does the method implemented by Perollo et al. to quantify immune cell infiltrates compare to previously validated approaches such as deconvolution methods, e.g. CIBERSORT [Newman et al., 2015 Nature Methods] or TIMER [Li et al., 2017 Cancer Research], and single-sample gene set enrichment analysis (ssGSEA) as in [Charoentong et al., 2017 Cell Reports].

5.d. From the heatmap it appears that only 5 genes (perhaps CD14, CSF1, CSF1R, CCL2, and CCL3) were used as IM signature. It is not clear why the authors ignored several other genes reported by Charoentong et al. as monocyte signature.

5.e. In panel a, was gene expression data standardized (i.e. z-scored) per row(gene)? If so, how come some rows/genes do not exhibit any gradient across the entire row? See top part of heatmap for Activated Dendritic Cells. Also, what does gray color indicate in the heatmap (see heatmap for M2 Macrophages, third row from top)?

5.f. In panel b, the authors are encouraged to replace the 3D plot with a 2D scatter plot where one axis shows CD14 score while the other axis shows hazard ratio (+/- 95% CI). The bases as well as the heights of the cones are difficult to read in the pseudo-3D format.

6. Figure 4

6.a. In panel a, microarray data should be deposited to public repositories such as GEO or Array Express, or be included as supplementary data.

6.b. How does the genes expression profile of selected sub-clones compare to human secretory subtype? See [Xu et al., 2014 Cancer Cell] for an example of such comparison.

6.c. In panel b, gene names are not readable.

7. Figure 5

7.a. In panel b, adjustment for multiple comparisons seems necessary.

7.b. It is not clear how the authors excluded the effect of CCR2 antagonist on other immune cells. For example, it has been shown that CCL2/CCR2 axis modulates macrophage polarization [Sierra-Filardi et al., 2014 J Immunol]. Others have shown that CCL2/CCR2 axis also plays a protective role by recruitment of gamma-delta T cells [Lanca et al., 2013 J Immunol]. Did the authors look at of M1/M2 TAMs or T cell subsets in response to CCR2 antagonist?

8. Figure 6

8.a. In panel a, how were the error bars computed with $n=2$?

8.b. In panel g, adjustment for multiple comparisons seems necessary.

8.c. Regarding panel h, how did the expression of F13A1 vary between LUSC subtypes in TCGA cohort? Also, it would be interesting to look at survival analysis in TCGA cohort based on F13A1.

8.d. Previous work has shown that Factor XIII activity is higher in advanced- vs. early-stage non-small cell lung cancer (NSCLC) patients [Lee et al., 2013 Yonsei Med J], and that Factor XIII is involved in tumor metastasis [Palumbo et al., 2008 J Thromb Haemost]. Porrello et al. take this further by associating Factor XIII to IMs. However, the data supporting this is somewhat limited in scope. Firstly, the authors do not provide sufficient evidence to exclude the contribution of other Factor XIII-expressing immune cells, namely tumor associated macrophages (TAMs), as previously suggested (e.g. see [Afik et al., 2013 JEM] and [Torocsik et al., 2005 Cell Mol Life Sci]). Secondly, the authors are encouraged to provide additional data demonstrating the effect of modulating Factor XIII in vivo.

General comments:

9. Supplementary methods on bioinformatics analysis is hard to follow and as a result difficult (if not impossible) to replicate in its current state. I recommend fully revising this section by splitting it into smaller subsections. Also, when using in-house algorithms, the authors are encouraged to provide their source code to facilitate reproducibility.

10. Throughout the manuscript, the authors refer to several non-significant results as “trends” to describe almost but not quite statistically significant results. This introduces ambiguity, may mislead the readers, and should be avoided (please see [Gibbs and Gibbs, 2015 BJA] for relevant discussion).

Reviewer #1 Comments:

The authors convincingly use patient data from the TCGA to highlight the elevated presence of CD14+ myeloid populations as a factor for poor survival in LUSC. The authors propose a mechanism whereby this cell population contributes to metastatic progression through Factor XIIIa. The study is interesting and presents new data in the field.

We thank the reviewer for these supportive comments.

1) The findings are somewhat surprising since it is more commonly thought that recruited monocytes differentiate into tumor-associated macrophages and MDSC, and these cells are more associated with pro-tumor activities. One possibility could be that only a subtype of CD14+CCR2+ myeloid cells, perhaps in a mid-differentiation state from inflammatory monocytes and mature macrophages/DCs, are associated with survival. Further characterization of the CD14+CCR2+ cells with additional markers could be necessary as they may not fit into the classical monocyte/M1/M2 classification.

This reviewer's comment is also similar to Reviewer #2, Comment #3c. To address this concern, we have developed a multiplex IHC protocol to stain M2 tumor-associated macrophages using a CD14/CCR2/CD206 triple stain. After staining 99 lung cancer tumors, we found that 85% of the cells (n=7,488) stained for CD14+CCR2+ alone, while only 15% of cells (n=1,342) stained for all 3 markers. We statistically confirmed that the percentage of CD14+CCR2+ cells of lung tumor samples that is also CD206 positive is a relatively small minority (p-value < 0.0001). This new data is shown in new Extended Data Figure 4. Although these findings support our use of CD14+CCR2+ to characterize inflammatory monocytes, the reviewer's comment highlights the need for more

sophisticated multiplex approaches to characterize immune populations. This is why our use of Immunogenomic analyses (Figure 3) are also complementary to our study of inflammatory monocytes, other CD14+ immune populations, and their overall relevance to lung squamous tumor biology.

2) The authors hypothesized that these myeloid cells could be enhancing the metastatic properties of the cancer cells. Although they compare secretion of TNF-alpha and CCL2 between the KLN205 and LN4K1 lines, they do not show that parental LN4K1 induce higher IM recruitment than parental KLN205 (only comparing with non-cancer cells). This data needs to be shown to validate the hypothesis that higher IM count leads to increased metastatic capabilities between the 2 cell lines. As the authors mentioned in the paper, other chemokines could also be responsible for recruitment. This concern is also supported by the fact that injection of CCL2-overexpressing KLN205 doesn't lead to a much higher IM count in the lungs compared to healthy mice (comparing Fig. 5e to Fig. 4j) or mice receiving CCL2-knockdown LN4K1 (Fig. 5e vs 5g), and yet leads to significant survival differences. Further, the recruitment of inflammatory immune cells to the lungs simply due to orthotopic injections should be addressed.

We appreciate the reviewer's comments and now show the direct comparison for the parental KLN205 and LN4K1 lines for IM counts and metastasis (Extended Data Fig. 11). This indeed demonstrates significant increases in IMs and metastases for the CCL2-high expressing LN4K1 sub-clone. Additionally, we have performed 'mock' HBSS injections in age-matched DBA2 mice and verified IMs are not increased over healthy age-matched DBA2 mice (Fig. 4j, left panel). Of note, the LN4K1 sub-clone also expresses increased CSF1 over the parental line (Fig. 4d), suggesting an additional mechanism for IM recruitment. This may in-part explain why CCL2 over-expression in KLN205 did not recruit as many IMs as the LN4K1 model. However, given our subsequent findings that CCL2 is necessary and sufficient to recapitulate LN4K1's metastatic phenotype, these findings imply that CCL2 may have other important biologic roles in IM's beyond its role in IM recruitment.

3) The in vivo experiments fail to show how IMs promote metastasis. Higher IM count could be independent from enhanced metastasis and could simply promote tumor initiation and growth, which will lead to enhanced metastasis. The following set of experiments also do not show a clear correlation between IMs and enhanced metastasis. Although the formation of invadopodia is shown across different conditions, migration and invasion assays are required to show the enhanced invasive capabilities of cancer cells under the various conditions. Since the authors show that IMs have a stronger correlation with poor survival than M2 macrophages (which is the polarization associated with TAMs), does FXIIIa expression decrease once differentiated to M2 similarly to resident macrophages? Authors should show that siRNA FXIIIa knockdown in IMs leads to diminished cancer cell invadopodia formation on Peak 1 fibrinogen; and that the T101 compound has no effect on invadopodia formation and invasion using a substrate other than fibrinogen.

We thank the reviewer for these comments as the resulting experiments have greatly improved the strength of the manuscript. In order to remain within the same co-culture conditions, we developed an assay in which the cancer cells were embedded in a fibrin clot above a migration chamber, and a chemoattractant was used to assess effects on cell invasion. Consistent with our invadopodia results, we found that cancer cells were markedly less invasive when grown in FXIIIa-deficient (Peak 1) fibrinogen, and invasion was rescued when co-cultured with inflammatory monocytes. The invasive phenotype rescue was completely abrogated in the presence of the Factor 13 inhibitor (T101) (Fig.

6h). Also, as suggested, we assessed this co-culture experiment when using a substrate (Matrigel) other than fibrinogen and found T101 had no effect on cell invasion (Extended Data Fig. 17). Similar to Reviewer #2, Comment #8d, we assessed the expression profiles of CD206^{High} (M2) and CD206^{Low} (M1) TAMs sorted from LN4K1 tumor-bearing mice. Compared with IMs, we found both TAM subsets had dramatically lower *F13a1* expression, similar to residential monocytes (Extended Data Figure 15). Additionally, we performed the invadopodia assay using IMs isolated from age-matched wild-type versus Factor XIIIa-null mice. Consistent with our assays with T101, we found that wild-type IMs had impressive capacity to rescue invadopodia, which was completely abolished when using FXIIIa-deficient IMs (Fig. 6i). Together, these findings strongly point toward FXIIIa-mediated IM biology being unique to IMs and less likely a shared feature of their differentiation into TAM subsets.

4) The authors do not show that blocking the Factor XIIIa-mediated mechanism that is proposed prevents metastasis in the orthotopic system *in vivo*.

We appreciate the reviewer's comment, which is similar to Reviewer #2, Comment #8d. A challenge in addressing this comment is that no LUSC metastasis models exist in a C57/B6 background (the only current strain with the FXIIIa^{-/-} context). To address this comment, we genetically modified THP1 monocytes to stably over-express (F13 ORF) or silence FXIIIa (F13 shRNAs 1+2) (Figure 7c). Using NSG mice to avoid rejection of THP1 and LN4K1 cells, we performed an experimental metastasis assay with 4 daily infusions of the respective THP1 cells (Figure 7d, schematic). One week later we necropsied the mice and enumerated micro-metastases. Consistent with our hypotheses, compared with LN4K1 injected mice, only the THP1-F13 ORF infused mice showed significant increases in metastasis, which was not observed in either THP1-F13 shR groups (Figure 7d).

5) Many of the phenotypic effects are of limited effect size. This could be addressed by incorporating more models to show the robustness of the differences in phenotypes in the different experiments.

We have taken several measures to address this reviewer's comment. First, we chose to incorporate an additional metastatic model of LUSC that we recently developed. We obtained the KAL cell line from Dr. Yinling Hu at the NCI, which was derived from a kinase-dead IKK α genetically-engineered mouse of LUSC (Xiao et al, *Cancer Cell*, 2013). We then performed the same *in vivo* selective strategy we used to isolate the LN4K1 subclone, which led to isolation of KAL-LN2E1 following 2 rounds of *in vivo* passaging. This cell line forms large, orthotopic LUSC tumors and rapidly develops lymph node and chest wall metastases (Extended Data Figure 12a+b). We then stably transduced the cells to silence CCL2 with various shRNAs (Extended Data Figure 12c). To assess whether IM recruitment is necessary for LUSC metastasis, independent of potential suppressor roles on T-cells (see Discussion about IMs v. mono-MDSCs), we injected these cells orthotopically into NSG mice. We then performed a cross-sectional necropsy on day 10 and annotated the effects of CCL2 knock-down on metastases. This revealed a significant reduction in the number and frequency of distant metastases, which was associated with reduced IMs in the primary tumor (Extended Data Figure 12d-f). We hope the reviewer can appreciate this was a substantial amount of work, further supports our initial findings and will add another invaluable LUSC metastasis model to the field. Also, as described above, we developed another co-culture model system in fibrinogen and demonstrated further roles of FXIIIa-expressing IMs to promote cancer cell invasion

(Figure 6h). We have also now demonstrated in a THP1 monocyte model that over-expression of FXIII A promotes lung squamous metastasis (Figure 7d).

Reviewer #2 Comments:

The manuscript by Porrello et al. aims to answer important questions regarding the heterogeneity of LUSC with respect to infiltration by immune cells, in particular myeloid cells, and highlights potential cell-intrinsic mechanisms associated with the observed heterogeneities. Overall, the manuscript is easy to follow and the findings are potentially relevant for development of targeted therapies in LUSC.

We thank the reviewer for the positive comments as well as the accompanying suggestions, which we believe have substantially strengthened the rigor of this work.

1) Figure 1

1.a. The authors have used the nearest shrunken centroid (NSC) classifier employed by [Wilkerson et al., 2010 Clin Cancer Res] to assign subtypes to TCGA tumors. The original classifier developed by Wilkerson et al. was based on microarray data, whereas TCGA data is based on RNA-seq. Like most machine learning algorithms, the NSC classifier operates under the assumption that the training and test data will be drawn from the same distribution. Challenges associated with cross-platform comparison of gene expression data (RNAseq vs. microarray [Thompson et al., 2016 PeerJ] or even between different microarray platforms [Shabaln et al., Bioinformatics 2008]) have been pointed out in the past. This raises doubt about the accuracy of subtype labels that Porrello et al. assigned to TCGA samples. The authors should provide evidence that the training and test data have similar distributions, or better yet apply the unsupervised consensus clustering approach of Wilkerson et al. to TCGA data to de novo determine transcriptional subtypes and compare their findings to that of Wilkerson et al.

We thank the reviewer for this comment. The reviewer states that our previously published lung squamous expression subtype classifier (Wilkerson CCR 2010) might perform differently on different platforms, specifically between gene expression microarray and RNA sequencing. The reviewer is correct that this classifier, and other similar classifiers (Sorlie et al 2003, Walter 2013 Plos One), have the assumption (in typical use cases) that cohorts are drawn from similar distributions, clinically and statistically. We agree with this point and have already published on this several times with our classifier.

In (Wilkerson et al, *Clinical Cancer Research*, 2010), we validated the classifier between 5 different microarray platforms by independent clustering and cross-validation. Please see the manuscript for further detail. Additionally, other groups have also directly used our classifier on new lung

squamous cell carcinoma cohorts (Brambilla et al, *Clinical Cancer Research*, 2015; Karlsson et al, *Clinical Cancer Research*, 2014; Clinical Lung Cancer Genome Project, Network Genomic Medicine. “A Genomics-Based Classification of Human Lung Tumors”, *Science Translational Medicine*, 2013) and validated the subtype classifications by expression pattern correspondence, independent clustering and incorporation of orthogonal feature associations. Regarding the reviewer's concern that our classifier might perform differently in microarray versus RNA sequencing platforms, we refer the reviewer to our prior publication in which we address this issue - see Supplementary Materials Figure S4.2, which we reproduce here, and Section 4 in The Cancer Genome Atlas Research Network, (2012) *Nature*. There, we applied our classifier to a new cohort of lung squamous cell carcinomas of the TCGA, which was assayed by mRNA sequencing on Illumina HiSeq (n=178) and of which a subset was assayed by Agilent 244K microarrays (n=122). First, the expression distributions of the validation gene sets and the subtype exemplar gene sets are highly concordant in a subtype-specific manner between Wilkerson 2010, TCGA RNA-seq and TCGA microarray cohorts. Second, the subtype classifications between TCGA RNA-seq and TCGA microarray were nearly identical; 94% of subtype calls agreed between the platforms as reported in the Nature publication. This 6% difference is similar to cross validation error rates of nearest centroid classifiers within the same cohort and platform. Therefore, our classifier performs consistently well on both microarray and RNA sequencing platforms (see figure insert).

To be completely responsive, we have generated an additional figure (Extended Data Figure 1) for this larger TCGA cohort similar to our prior Nature publication. Here, we display the Wilkerson 2010 cohort expression for the validation gene set, and the marker gene set. The TCGA cohort (n=491) is displayed in the same gene order. Both the validation gene set and the marker gene set again show highly concordant expression subtype-specific patterns between both cohorts. Therefore, our published classifier continues to be robust across platforms including RNA sequencing and our subtype labels in the current manuscript are highly accurate.

1.b. In panels a, b, and d, how did the authors address multiple testing?

With these three panels we offer the Reader three alternative levels of analysis of the survival difference of LUSC patients, focusing on distinct relationships between LUSC subtypes and survival. In panel (a) we compare the four LUSC subtypes, all at the same time, using the log-rank (Mantel-Cox) test (GraphPad Prism), which is based on a chi-square statistic for four groups of patients (samples), with three degrees of freedom. This test is treated as unique and, for this reason, no multiple hypothesis testing correction (MHTC) was made; the main statistical output is the p-value itself, which equals 0.127. In panel (b) we wished to understand if any subtype(s) is (are) substantially different from the others in terms of survival: strictly speaking, no case can be made for the statistical significance (log-rank test) of any of them in the individual analysis (all p-values are > 0.05). We performed an MHTC on the four p-values of panel (b), as requested by the Reviewer, and calculated the corresponding false discovery rate (FDR) according to Benjamini and Hochberg (linear step-up procedure). The revised panel (b) includes these values (0.2448, 0.3939, 0.3939, 0.2240 for Classical, Basal, Primitive, and Secretory, respectively). These results confirm that there is no statistically significant difference among them (making this outcome clearer for Secretory). Finally, in panel (d), we performed a direct comparison (log-rank test) between Classical and Secretory, due to their degree of ‘genomic dissimilarity’, as for RNA-Seq, which can be visually appreciated looking at Fig. 1c (and was numerically defined in the revised Supplementary

Methods). This genomic dissimilarity is the largest among couples of LUSC subtypes, and we wished to check if this outcome was matched by a difference in survival; indeed, the log-rank p-value is <0.05 . Since this test is used for evaluating a distinct type of hypothesis and was used only once, no MHTC was performed. In summary, we took these actions in the revised manuscript: 1) we replaced the p-values with the FDRs in panel b; 2) we explicitly mention, in the revised Supplementary Methods, the computational motivation (genomic dissimilarity) that brought us to perform the log-rank test shown in Fig. 1d; 3) most importantly, we have modified the first paragraph of the Results, clarifying our view about the survival curves of panels a, b and d and removing any reference to their statistical significance. Indeed, these panels mostly have a descriptive value and help to revise the survival differences of TCGA LUSC subtypes (see Wilkerson et al, 2010 *Clinical Cancer Research* and The Cancer Genome Atlas Research Network. Comprehensive genomic characterization of squamous cell lung cancers. *Nature*, 2012); which are also used as a benchmark for further analyses. Because these results were negative or statistically just below the level of statistical significance (depending on the chosen statistical approach), we assessed also more specific agents potentially involved in survival, such as genes and immune cell types.

1.c. In panel c, it is not clear to me how the 4291 genes shown in the heatmap were identified. From the supplementary methods, it appears that the authors first performed some filtering based on variability, and then performed differential expression analysis. However, it is not clear how the authors addressed the mean-variance trend inherent to RNAseq data during variance filtering and differential expression analysis.

In this manuscript, we have utilized a combination of computational approaches, which we consider complementary and interlocking. As described in the paragraph 'Analysis of RNA-Seq data: identification of a set of representative genes' of the revised Supplementary Methods, the gene selection criteria of Figure 1c, which generate a hybrid differential analysis, are based on a) evidence of difference between the Classical-Basal and Primitive-Secretory subtypes (ratios of the medians), b) statistical significance of this difference, c) signal quality as measured by the median and 'presence' level, and d) (subtype-independent) gene heterogeneity, as measured by the standard deviation. This approach aims to reasonably balance the selected genes from several standpoints, setting some priorities among these requirements and including a priori information. Notably, all chosen thresholds are either intrinsic to the data (such as, for instance, the standard deviation and the ratios of the medians required to belong to the top 50% across the whole set of genes) or canonically used (such as p- and q-value < 0.01). Using this approach, the direct penalization for genes with low standard deviation (and, of course, low variance) is greater than for low median (and, generally, mean), because the priority, with respect to these two variables, was discarding genes that are homogeneously expressed across the samples of the four subtypes. Interestingly, even the strong requirement that we impose on the standard deviation would not be able, alone, to exclude some genes that are mostly 'null', so the chosen multiple requirements create a strong filter against (very) low quality genes. As for the indirectly generated selection against genes that have lower mean due to the mean-variance trend, we accepted it as part of our hybrid selection (where the gene heterogeneity plays an important role) and statistical approach. Indeed, differences across the subtypes were measured using the non-parametric Wilcoxon rank sum test, so that means and variances are not directly involved and linear models are not invoked or dealt with for hypothesis testing. With these steps, we laid the ground for more specific analyses and for the experimental validation of our computational findings.

Overall, the gene list of Figure 1c aimed to give an overview of the data and to collect high-quality genes with good statistical properties for other analyses described in the manuscript, such as i) the survival analysis based on gene expression levels, ii) Ingenuity Pathway Analysis (IPA) for ‘Disease or Function’ and IPA ‘Upstream Regulator Analysis’, and iii) gene ontology (GO), which is typically limited in the number of genes concurrently used (for DAVID, for instance, the suggested (tested) value is 3,000 and, indeed, after a further refinement, both the upper and lower portion of this heat map, which underlie distinct biological features, were kept below this limit). However, we also exploited more comprehensive approaches, such as GSEA, because we wanted to have multiple types of results, relying on diverse selection thresholds, to confirm that the computationally identified biological themes or genes were biologically important. Once it became clear that the immune response was paramount from every angle we looked at, we analyzed the immune infiltrate of these tumors, where all genes that have been described for a specific immune cell type are included, regardless of i) their mean and standard deviation/variance, and ii) their statistical significance with respect to the differences among subtypes.

1.d. What does “hyper-expressed genes” mean? Is it synonymous to “significantly up-regulated genes”? If so, please stick to the latter to avoid confusion.

We apologize for the confusion generated by this word, which we fixed in the main text and Supplementary Methods, since it had been used with different meanings in these two parts. In the main text, this word was replaced by the phrase ‘differentially-expressed’ (an outcome that, as explained in the revised ‘Supplementary Methods’, is joined with statistical significance for the displayed genes). Instead, the adjective ‘hyper-expressed’ used in the ‘Supplementary Methods’ was not synonymous with ‘significantly up-regulated’; so, we maintained this word, but explained its precise meaning. Specifically, a sample *s* hyper-expresses a gene *g* when the expression of *g* in *s* is \geq the average expression of *g* in the whole cohort of patients. We defined what we currently intend with hyper-expression in the new paragraph ‘Analysis of RNA-Seq data: batch survival analysis’ and, when the same adjective is used (in two other parts of the revised Supplementary Methods), we refer the Reader to that definition, to avoid any ambiguity. Additionally, we added a sentence in our Supplementary Methods (in the new paragraph ‘Gene Ontology Analysis of RNA-Seq data’) that talks about genes significantly up-regulated, to better address this Reviewer’s concern,

1.e. Regarding panel h, in the main text it is stated that several markers of IMs were leading predictors of progression (i.e. had large hazard ratios), however only 2 markers are listed in the figure. A more objective approach would be to compare the location of IM- and non-IM-associated genes on a ranked list of survival genes.

We would like to point out that 7 of the 9 markers in Figure 1h are related to inflammatory monocytes. However, the top 2 markers in red represent those with hazard ratio p-values ≤ 0.001 . As suggested, we have updated Extended Data Table 2, which is ranked according to hazard ratio and includes a column denoting whether genes are IM- and non-IM-associated genes. This demonstrates that 10 of the total 403 genes with significant hazard ratios are markers of inflammatory monocytes.

2) Extended Figure 1

2.a. In the main text, it is stated that LUSC subtypes were balanced for clinical features, whereas Wilkerson et al. have reported that tumor differentiation and patient gender were

associated with subtype. Did the authors check patient gender as part of clinical features in TCGA data? What's the possible explanation for this potential discrepancy?

We had checked the gender of these patients; in fact, a clinical feature summary can be found at the bottom of Extended Data Table 1. As for the gender, we observed that: 1) the number of female subjects is considerably lower than the number of male subjects (24.71% vs. 75.29%); 2) in no LUSC subtype the statistical significance is reached when performing a binomial test and comparing the number of male and female patients, while also accounting for this difference in frequency between the two sexes. Therefore, the reason why in each subtype there are more males than females is because the same happens in the general population sampled, without a strong subtype-based characterization; 3) the only partial (not statistically significant) exception, is found in the Classical subtype, where the percentage of females drops to 18.03%, fairly below the level found in the whole population sampled (24.71%), which corresponds to a p-value of approximately 0.09 (however, this outcome becomes less interesting looking at the corresponding FDR). When the four subtypes are assessed looking for a general unbalance between the two genders across the four subtypes, using the statistical test used in the article of Wilkerson et al. (Fisher's exact test) the p-value is also not statistically significant. Therefore, focusing on the statistical significance, these results are mostly (although not completely) consistent with what was calculated by Wilkerson et al., considering that in that article (based on several cohorts of patients) it was reported that: 1) the p-value, for this clinical variable, approaches the statistical significance, but without reaching it (this outcome is described more explicitly in the Results than in the Abstract); 2) males are over-represented in the Classical subtype. Finally, the tumor grade was not available in the files downloaded from the TCGA Data Portal and Firehose data hub, so we could not compare this clinical variable with what was published in our prior report by Wilkerson et al.

3) Figure 2

3.a. In panels a, b, and c, how did the authors address multiple testing?

Panel (a) of Figure 2 contains a survival comparison between high and low levels of CD14 (plot on the left) and a survival comparison of patients according to their quartiles of CD14 expression (plot on the right). Notably, the right plot of Fig. 2a is generated using a unique log-rank (Mantel-Cox) test (GraphPad Prism), which is based on a chi-square statistic with three degrees of freedom, since there are 4 groups (samples) displayed and, as such, does not require MHTC by itself. Both tests are meant to clarify the role played by CD14 and, because these two analyses split the same samples in different ways (the second being a refinement of the first) we had deemed that a specific MHTC vs. each other was not necessary. To allow the simultaneous assessment of these two hypotheses, we have conservatively estimated the p-value in the left plot as 0.0001 (thus being consistent with our previous manuscript), and calculated their FDRs, which are now reported in this panel (0.0002 (left) and 0.0006 (right)). As for panel (b), we recalculated the shown p-values (for computational consistency with some new parts) and derived the corresponding FDRs to make sure that also at this level the statistical significance is reached. The four FDRs are: 0.0019, 0.4505, 1.6342e-04, 1.7044e-09, respectively, from left to right. Since the general statistical significance is essentially unchanged after MHTC and because of word count limits as well, we omitted to show these four values in the figure legend. Finally, we performed the MHTC of the three cytokines of Fig. 2c, namely CCL2, CCL3, CSF1, with respect to the median as splitting threshold. The corresponding FDRs are: 0.0036, 0.0177 and 0.0177. For Fig. 2a and 2c, we

report the FDRs in the panel and the p-values in the legend, to allow the Readers to evaluate the MHTC also independently, if desired.

3.b. In panels d, e, and f, what is the unit of y axes (gene expression)? RSEM expected or normalized counts? Did the authors performed statistics on counts or log-transformed counts?

The units are RSEM normalized counts, whose values were obtained from the TCGA Data portal (rsem.genes.normalized_results) and that are matched by the Firehose Broad GDAC data hub values (RSEM_genes_normalized_data), to keep our analyses consistent with one of the main TCGA measures of gene expression.

As for panel d: a) the Pearson's correlation coefficient (ρ) was calculated, as customary, on non-transformed values (in particular, standing clear of non-linear data transformations); b) the shown p-values are based on a t-test statistic with 346 (= number of samples (348) - 2) degrees of freedom; 3) the three Spearman's rank correlation coefficients confirmed and reinforced what was shown by the Pearson's coefficients, being higher than the three ρ s of this panel. We have described these statistical analyses in the revised Supplementary Methods.

For panels e and f, we used the ANOVA test based on its robustness, as for the normality requirement (see, for instance, E. Schmider, M. Ziegler, E. Danay, L. Beyer, M. Bühner. Is it really robust? Reinvestigating the robustness of ANOVA against violations of the normal distribution assumption. *Methodology* 2010; Vol. 6(4): 147-151), selecting log-transformed data (while being plotted in "natural" units). P-values are expressed as < 0.0001 and, interestingly, this annotation holds both using non-transformed and log-transformed data, since p-values are largely below this threshold for both types of data. As for the homoscedasticity requirement, we observed that, after log-transformation: 1) CCL3, CSF1, and CD14 pass a) the (conservative) Bartlett's test (p-value threshold: 0.05), b) the Brown-Forsythe test (p-value threshold: 0.05); c) the criterion about the ratio between the maximum and minimum variance among the four subtypes proposed by Dean, Voss and Draguljić [Dean, Voss, and Draguljić. *Design and Analysis of Experiments*, 2017, Springer, 2nd Edition, pp.110-111]; 2) CCL2 passes both the Brown-Forsythe test and the aforementioned criterion of Dean, Voss and Draguljić. Notably, these ANOVA results were confirmed also using the Kruskal-Wallis test. Altogether, the four ANOVA tests were deemed reliable and their p-values pointed towards a remarkable difference among the subtypes for these four genes. For these reasons, the p-values shown in the original manuscript (that are expressed conservatively) were not changed in the revised manuscript. Some of these statistical steps, which were previously omitted, have been described in the revised Supplementary Methods, compatibly with the need to limit the size of this section.

Finally, we had treated differently the ANOVA of TNF α (Extended Data Figure 9), so we have modified that analysis (whose results are statistically significant also in the new version). Also for TNF α the three afore-mentioned criteria for homoscedasticity were met and results were confirmed through the Kruskal-Wallis test.

We sincerely thank the Reviewer for these questions, which allowed us to fill these methodological gaps in our manuscript.

3.c. In panels g, h, and i, how are the authors confident that CD14+/CCR2+ cells are IMs and not TAMs?

We appreciate the reviewer's comment, which is similar to Reviewer #1, Comment #1. To address this concern, we developed a multiplex IHC protocol to stain M2 tumor-associated macrophages using a CD14/CCR2/CD206 triple stain. After staining 99 lung

tumors, we found that 85% of the cells (n=7,488) stained for CD14+/CCR2+ alone, while only 15% of cells (n=1,342) stained for all 3 markers (Extended Data Figure 4). We statistically confirmed that the percentage of CD14+/CCR2+ cells of lung squamous tumor samples that is also CD206 positive is a relatively small minority (p-value < 0.0001). We believe that these new data support the use of CD14+/CCR2+ as markers of inflammatory monocytes as shown in Figures 2g-i.

4) Extended Figure 2

4.a. Adjustment for multiple comparisons seems necessary.

The FDRs of the three columns of this plot (comparisons of Basal/Primitive/Secretory vs. Classical), which are, respectively, 0.1598, 0.2654, and 0.1110, are now shown in this figure legend.

5) Figure 3

5.a. The analysis performed is strongly dependent on gene signatures. The authors should provide complete list of gene signatures used in their analysis as supplementary material.

As suggested we have now provided the list of gene signatures used in the analyses in Extended Data Table 7.

5.b. Several studies (including those cited by Porrello et al.) have provided objectively selected gene signatures. In the supplementary methods, the authors have stated that the gene lists were made comprehensive by adding selected markers. The addition of CD14 (which is not a monocyte-specific marker), CSF1 and CSF1R (which are canonical TAM and DC markers), as well as cytokines of interest (CCL2/CCL3) to the IM signature seems to introduce bias in the results given that as shown in Figure 2 these genes are expected to be overexpressed in the secretory subtype.

Because there are currently no gene signatures for human inflammatory monocytes, we curated our gene list from the literature describing the most validated cell surface markers (CD14, CCR2, CSF1R) and recruitment chemokines (CCL2, CCL3 and CSF1) used to characterize these cells. As stated below in response to 5d., the Charoentong *et al* “monocyte” signature did not distinguish between classical “inflammatory” and residential “patrolling” monocytes, which are highly divergent in gene expression and function, and thus we did not use this signature in our model. As pointed out by the reviewer, a growing challenge in the field of immunology is the fact that some markers are also expressed on other immune populations (e.g. CCR2 and CD14 are also part of the MDSC signature). Additionally, several markers represent a continuum of cell differentiation (e.g. CSF1R remains on tumor-associated macrophages as IMs differentiate). It was due to the fact that other immune cells may express CD14 (albeit at lower levels) that we chose to perform the analyses shown in Figure 3 to understand the relative contribution of CD14+ populations and survival. Our findings that activated dendritic cells and M2 macrophages (both derivatives of inflammatory monocytes) scored the #2 and #3 most significant hazard ratios (Figure 3b+c) further support the robustness of the IM signature and importance of these ‘precursor’ cells. While use of immunogenomics and multiplexed IHC are powerful new tools, there are still many caveats to their usage and the immune populations of interest requires validation in experimental models. We believe our subsequent validation work as outlined in this letter, and our characterization of FXIIIa’s role in inflammatory monocytes (the

expression of which highly correlated with our IM signature, Extended Data Fig. 16a) further support the use of our IM signature.

5.c. How does the method implemented by Porrello et al. to quantify immune cell infiltrates compare to previously validated approaches such as deconvolution methods, e.g. CIBERSORT [Newman et al., 2015 Nature Methods] or TIMER [Li et al., 2017 Cancer Research], and single-sample gene set enrichment analysis (ssGSEA) as in [Charoentong et al., 2017 Cell Reports].

When we decided how to analyze these data, we looked for a straightforward computational approach based on what had already been published in this field and as robust as possible. Our method is directly derived by the algorithm of Iglesia et al. [Iglesia, MD et al. J Natl Cancer Inst., 2016], which had been applied to quantify several immune cell infiltrates in cancer samples. An important difference with that algorithm is that genes do not contribute to the immune signatures with their expression, but rather with their expression ranks (across the available samples). This change prevents markers/genes from being over- or under-estimated in that signature, removing the biases produced by highly and lowly expressed genes. Notably, our method does not aim to perform any deconvolution of RNA mixtures (like CIBERSORT and TIMER), which involves the estimation of the relative proportions of the cell types of interest. Similarly, it was not our goal to define if and how much the picked sets of immune-related genes were over- or under-represented in each sample, relatively to each other, like it would happen using ssGSEA. For instance, while for ssGSEA a gene set whose genes are lowly expressed in a sample s is impoverished in s , for our method the same gene set could be, potentially, even enriched in s , if its genes happen to be more highly expressed in s than in most of the other samples. Altogether, our algorithm is focused in ‘horizontally’ evaluating each gene set across the TCGA LUSC samples i) without making assumptions of greater importance for a gene set rather than another because its genes are expressed to a greater extent, ii) without implicitly imposing limitations about how many cell types are going to be assessed as highly or lowly expressed, iii) with reduced mutual interferences among the assessed cell types (i.e., with maximized independence among the assessed immune gene signatures), iv) creating an effective filter against genes with many missing values (null or almost null genes), due to the way in which our cell-type-specific sample score works, v) helping to define a sample spectrum (from immunologically ‘cold’ to ‘hot’) with respect to each immune cell type, and vi) providing an output that can be easily used for downstream statistical analyses (see the Supplementary Methods).

Additionally, we would like to point out that while deconvolution is a topic distinct from ours (somehow complementary), it also clearly introduces a remarkable degree of complexity in the analysis of the tumor infiltrate, which will likely be sorted out in the future, when more computational and statistical research will be published in this field. About this subject see, for instance, the Introduction of the above-referenced paper (introducing TIMER) of Li et al., which mentions statistical concerns about CIBERSORT multicollinearity, a paper from Li et al. [Li et al., Genome Biology 2017] and the discussion of the manuscript of AM Newman et al. about CIBERSORT itself. Newman et al. [Newman et al., Genome Biology, 2017] have recently commented the differences between these two methods and their discordant results.

Since we think that this Reviewer’s topic is of interest to a number of Readers, we have clarified our computational goals concerning the LUSC infiltrates in the main text and modified our Supplementary Methods, which now briefly illustrate the differences between our method and these three.

5.d. From the heatmap it appears that only 5 genes (perhaps CD14, CSF1, CSF1R, CCL2, and CCL3) were used as IM signature. It is not clear why the authors ignored several other genes reported by Charoentong et al. as monocyte signature.

The IM signature contains 6 genes, however CD14 is placed at the top to show how it corresponds with other relevant functional genes (*novel IM gene*: F13a1, *classic tumor-associated macrophage gene*: VEGFa, *classic MDSC genes*: Arg1, NOS2). We have now stated this in the figure legend. We did not use the Charoentong et al monocyte signature because it is too generic and does not distinguish between inflammatory and residential monocytes. Given that these cell types (IMs v. RMs) have divergent functions and gene expression profiles, we felt it was not a well-suited signature to apply to our analyses.

5.e. In panel a, was gene expression data standardized (i.e. z-scored) per row(gene)? If so, how come some rows/genes do not exhibit any gradient across the entire row? See top part of heatmap for Activated Dendritic Cells. Also, what does gray color indicate in the heatmap (see heatmap for M2 Macrophages, third row from top)?

Some gene expression measures, according to the LUSC TCGA data, are null; therefore, when Cluster 3.0 performs its pre-processing steps, these values are log2-transformed (this is the first step of Cluster 3.0 pre-processing) and become, computationally speaking, Not-a-Number (NaN). The gene centering (data standardization) always follows, in our analyses, this pre-processing step. Treeview, afterwards, displays these numbers as missing values, using the gray color for representing them. While there would be ways to computationally go around this issue (such as using numerical offsetting), we prefer to maintain this data feature, so that mRNAs that are very lowly expressed/absent are easily distinguishable from the others. To clarify this point, we implemented the heat map legends of Fig. 3a and Extended Data Fig. 5 adding a gray rectangle (that points to 'null normalized values'), which illustrates this color-coding feature. We have modified the Supplementary Methods as well, to explain this feature and avoid any ambiguity. As for the color gradient, the samples of this figure were ordered according to growing levels of CD14, which means that, unless there is a (positive or negative) correlation between CD14 and the displayed gene, the order of green and red rectangles for that heat map row may resemble a random sequence of these two colors. We also want to highlight that, because the color black is used to represent samples having an expression near to the median (see the heat map legend), gene gradients are more or less visible, depending on the number of samples that are close to the gene median. This heat map feature too is, in our opinion, helpful, since the Reader can quickly identify genes whose probability density functions are more centered around their median and genes that have, instead, a broader dynamic range. Finally, it may be worth saying that the color coding that we chose is important for visualizing these data in a way that delivers as much information as possible just using heat maps, but is independent of the algorithms that we used to analyze the immune cell types.

5.f. In panel b, the authors are encouraged to replace the 3D plot with a 2D scatter plot where one axis shows CD14 score while the other axis shows hazard ratio (+/- 95% CI). The bases as well as the heights of the cones are difficult to read in the pseudo-3D format.

As suggested we have made a 2D scatter plot showing the CD14 score (x-axis) and hazard ratio (y-axis, including +/- 95% CI), now in revised Figure 3b. We have removed the 3D plot.

6) Figure 4

6.a. In panel a, microarray data should be deposited to public repositories such as GEO or Array Express, or be included as supplementary data.

We have added a table (Extended Data Table 8) with the numerical results of the comparisons i) LN4K1 vs. KLN205 and ii) LN4K1 vs. MBEC samples, which possibly shows most of the information desired by the Reviewer, since it is referred to the entire heat map of Figure 4a. Additionally, we are completely ready, according to the guidelines of Nature Communications and the Editor's directions, to deposit the raw data of the samples of our murine lung squamous carcinoma model in the Gene Expression Omnibus (GEO) public repository or to include them as supplementary data.

6.b. How does the genes expression profile of selected sub-clones compare to human secretory subtype? See [Xu et al., 2014 Cancer Cell] for an example of such comparison.

We gladly performed this analysis: it is now described in the Supplementary Methods, and these results have been included in Extended Data Figure 10, together with a reference pointing to this scientific article. Altogether: 1) our gene selection algorithm has been balanced between what was described by Xu et al. and what we had done throughout our manuscript, with the goal not to confuse the Readers and keep heat maps and analyses comparable and consistent; 2) we avoided overly complex and lengthy analyses. Indeed, they would be unavoidably divergent from what was described by Xu et al., due to the presence of more groups of samples both for our murine (KLN205 cell line) and human (TCGA LUSC data); 3) we focused exclusively i) on the two mouse group samples at the extreme of the heat map of Fig. 4a (i.e., the normal murine bronchial epithelial cells (MBEC) and the sub-clone LN4K1), treating KLN205 as a 'transition state' between these two groups, and ii) on the two LUSC subtypes with the greatest genomic dissimilarity (see our answer above and the revised Supplementary Methods), i.e., Classical and Secretory; 4) we evaluated that, with respect to Xu et al., this mixed comparison may have more nuances and be harder to interpret. Indeed, the difference between MBEC and LN4K1 in mouse reflects the difference normal vs. metastatic cell, while the Classical-Secretory comparison in LUSC partially mirrors the transition between immunologically 'cold' and 'hot' samples and also correlates with a higher hazard rate. For this reason, we decided to be stringent in our gene selection, so to make the consequential GO analysis as robust and enlightening as possible; 5) at the same time, we did not take this stringency to the extreme, to collect enough genes for the GO analysis; 6) the GO threshold of statistical significance was fixed higher than we usually would do, due to the presence of these confounding factors; 7) we matched MBEC and Classical on the one side and LN4K1 and Secretory on the other, for biological consistency, bearing in mind the supervised approach used by Xu et al. in defining the matched groups; 8) the heat maps that we show side by side have, in every row, orthologous genes, and the two possible directionalities (ratio of the medians above and below 1) are separately displayed; additionally, the visualization style chosen for this analysis was consistent with the other parts of this manuscript; 9) taking into account the biological context, we used this analysis for understanding which genes (and gene ontology terms) are, at the same time, independent of the micro-environment and shared between our metastatic mouse model of LUSC and LUSC patients who generally have a worse prognosis and also have higher density levels of most immune cell types. One interesting outcome of the GO analysis was that shared genes, which are more highly expressed in Secretory, contribute to the inflammatory response (p -value = 4.18×10^{-4}) and include $TNF\alpha$ and CCL2. Overall, comparing these two results helps to understand

some features of the microenvironment, but should be done carefully, because of the large difference in size between the gene sets used here and in our main GO analysis (see Fig. 1c and 1f). In summary: 1) many immuno-related biological processes of the GO shown in Figure 1 are missing from the new analysis (shown in Extended Data Figure 10); many of which are undoubtedly immune microenvironment-dependent (and thus missing from our cell line microarray). In this sense, we can appreciate the complexity of the immune-related GO terms of Secretary vs. this new GO list; 2) genes of the inflammatory response emerge at least in part through a microenvironment-independent context (e.g. tumor cell derived TNF α and CCL2). We hope that the Reviewer will find interesting these computational choices and the new results.

6.c. In panel b, gene names are not readable.

We now provide an Extended Data Table 9 for full gene name reference and columns detailing their molecular type and p-value for overlapping function.

7) Figure 5

7.a. In panel b, adjustment for multiple comparisons seems necessary.

To improve the statistical analysis of this panel, we have aggregated the four t-tests related to the quantification of the luciferase signal of our mouse model and used these p-values for calculating the corresponding FDRs (according to Benjamini and Hochberg). After this MHTC, the statistical landscape is not much changed, because the four FDR are, respectively: a) 0.0313 (KLN205-Scr ORF vs. KLN205-CCL2 ORF), b) 0.4187 (LN4K1-Cntrl shR vs. KLN205-CCL2 ORF), c) 0.0034 (LN4K1-Cntrl shR vs. LN4K1-CCL2 shR#1), and d) 0.0034 (LN4K1-Cntrl shR vs. LN4K1-CCL2 shR#2). To display these statistical results, we have switched cases c and d to two asterisks, since they are associated with FDR < 0.01, and have clarified this in the figure legend.

7.b. It is not clear how the authors excluded the effect of CCR2 antagonist on other immune cells. For example, it has been shown that CCL2/CCR2 axis modulates macrophage polarization [Sierra-Filardi et al., 2014 J Immunol]. Others have shown that CCL2/CCR2 axis also plays a protective role by recruitment of gamma-delta T cells [Lanca et al., 2013 J Immunol]. Did the authors look at of M1/M2 TAMs or T cell subsets in response to CCR2 antagonist?

We have now performed additional experiments with our LN4K1 model and assessed the effects of PF-04136309 (CCR2i) on macrophage polarization and gamma-delta T-cells. To be consistent with our prior experiments, we assessed these immune populations following 1 week of treatment. We chose this treatment scheme since it was significantly associated with reductions in inflammatory monocytes and metastases (Figure 5h-k). We did not observe any reductions in CD206^{Low} (M1) or CD206^{High} (M2) TAMs (Extended Data Fig. 14a). This is consistent with a prior report of this compound (Mitchem et al, *Cancer Research*, 2013) that found IMs were depleted in 4 days but it took at least 8 days for TAM depletion. We also did not see any reductions in gamma-delta T-cells (Extended Data Fig. 14b), nor did we see any effects on CD8, CD4 or regulatory T-cells (Figure 5l). We thank the reviewer for this comment, and we believe these newer data further support that the therapeutic effects seen with CCR2i are from depletion of IMs.

8) Figure 6

8.a. In panel a, how were the error bars computed with n=2?

We appreciate the reviewer's comment and now realize the way we originally presented these data were unclear. Also, inclusion of VEGFa expression did not fit well into the context of the role of FXIIIa expression in inflammatory monocytes. We now present relative expression between inflammatory and residential monocytes for *F13a1* in 3 mice (each shown separately, Figure 6a). The error bars here represent technical triplicates for qPCR for each of the biological samples.

8.b. In panel g, adjustment for multiple comparisons seems necessary.

As similarly done in Figure 5b, we have aggregated the five t-tests of Figure 6g as part of the experiments to understand the relationships between F13A1 and inflammatory monocytes. For consistency with our previously submitted manuscript, we have used the same p-values and, conservatively, since each of them was expressed as <0.0001 (with three asterisks for their statistical significance), we have rounded them to 0.0001 for MHTC purposes. The Benjamini-Hochberg FDR of these statistical tests is, for all five cases, 0.0001. Because this MHTC does not substantially change our assessment about the statistical significance of these comparisons, we have kept this panel as it is, while mentioning the FDR of these five tests in the legend of Figure 6.

8.c. Regarding panel h, how did the expression of F13A1 vary between LUSC subtypes in TCGA cohort? Also, it would be interesting to look at survival analysis in TCGA cohort based on F13A1.

We performed these analyses, which now are shown in Extended Data Figure 16 and Figure 7, respectively. Remarkably, the distribution of values above and below median across the four subtypes for F13A1 is consistent with what we had found for CD14. In statistical terms: a) Classical has a binomial test p-value of 0.3193 and a FDR of 0.4257; b) Basal has a binomial test p-value as well as FDR of 0.9142; c) Primitive has a binomial test p-value of 0.02701 and a FDR of 0.0540; d) Secretory has a binomial test p-value of 0.005014 and a FDR of 0.0201. Therefore, when looking at the FDR, the only subtype for which there is statistical significance is Secretory. Also, the levels of F13A1 are directly related with the survival of LUSC patients, resembling what was found for CD14 (higher F13A1 levels are associated with a worse survival rate), although with a higher p-value (log rank p-value = 0.0084).

8.d. Previous work has shown that Factor XIII activity is higher in advanced- vs. early-stage non-small cell lung cancer (NSCLC) patients [Lee et al., 2013 Yonsei Med J], and that Factor XIII is involved in tumor metastasis [Palumbo et al., 2008 J Thromb Haemost]. Porrello et al. take this further by associating Factor XIIIa to IMs. However, the data supporting this is somewhat limited in scope. Firstly, the authors do not provide sufficient evidence to exclude the contribution of other Factor XIII-expressing immune cells, namely tumor associated macrophages (TAMs), as previously suggested (e.g. see [Afik et al., 2013 JEM] and [Torocsik et al., 2005 Cell Mol Life Sci]). Secondly, the authors are encouraged to provide additional data demonstrating the effect of modulating Factor XIIIa in vivo.

We appreciate the reviewer's comments, which are similar to Reviewer #1, Comments #3+4, and we have taken several experimental approaches to address them. As suggested, we have FACS sorted IMs, RMs, M1 (CD206^{Low}) and M2 (CD206^{High}) tumor-associated macrophages and assessed relative *F13a1* expression by qPCR. These data revealed that Factor 13 expression is markedly increased in IMs as compared not only with RMs but also both M1 and M2 TAM subsets as well (Extended Data Figure 15). To

address whether FXIIIa-expressing monocytes are sufficient to promote LUSC metastasis, we genetically modified THP1 monocytes to stably over-express (F13 ORF) or silence FXIIIa (F13 shRNAs 1+2) (Figure 7c). Using NSG mice to avoid rejection of THP1 and LN4K1 cells, we performed an experimental metastasis assay with daily infusions of the respective THP1 cells (Figure 7d, schematic). One week later we necropsied the mice and enumerated micro-metastases. Consistent with our hypotheses, compared with LN4K1 injected mice, only the THP1-F13 ORF infused mice showed significant increases in metastasis, which was not observed in either THP1-F13 shR infused groups (Figure 7d).

9) Supplementary methods on bioinformatics analysis is hard to follow and as a result difficult (if not impossible) to replicate in its current state. I recommend fully revising this section by splitting it into smaller subsections. Also, when using in-house algorithms, the authors are encouraged to provide their source code to facilitate reproducibility.

This section was extensively revised and broken down into smaller parts to allow reproducing our results and make it more readable. Indeed, the number of these paragraphs has been doubled (from 9 to 20, some of which are completely new). Several sentences were made shorter and clearer, while parts that we had previously omitted were included to avoid ambiguity. In the parts where we mentioned the use of in-house scripts, we addressed this Reviewer's request. Precisely, where we talk about gene-based survival analysis, we provide a MATLAB script that shows how it is performed. As for the part of the microarray data analysis, considering the existence of different annotation files from Affymetrix and in order to grant portability, we have directly provided the file with the probe annotation of the selected coding genes of this platform (Extended Data Table 12).

10) Throughout the manuscript, the authors refer to several non-significant results as "trends" to describe almost but not quite statistically significant results. This introduces ambiguity, may mislead the readers, and should be avoided (please see [Gibbs and Gibbs, 2015 BJA] for relevant discussion).

As suggested, when results are non-significant we have modified the text to state this and in-turn have removed use of the term "trends" to avoid any such ambiguity.

In summary, we were very pleased to note that the reviewers found our work interesting, and important, and we appreciate their constructive suggestions for extending the scope and detail of these studies. Thank you in advance for your thoughtful consideration. Please let me know if I can provide any further information.

REVIEWERS' COMMENTS:

Reviewer #1 (Remarks to the Author):

The authors have addressed my concerns.

Reviewer #2 (Remarks to the Author):

Porrello et al. have substantially improved their manuscript by providing new computational and experimental data. Results are clear, and the manuscript is well written. The authors have thoroughly addressed all the points that I raised earlier. I'm very pleased by the quality and depth of the author's response.